# Fine-tuning can cripple your foundation model; preserving features may be the solution

**Jishnu Mukhoti**                                                                 *jishnu@robots.ox.ac.uk*
*Department of Engineering Science, University of Oxford*

**Yarin Gal**                                                                       *yarin.gal@cs.ox.ac.uk*
*Department of Computer Science, University of Oxford*

**Philip H.S. Torr**                                                             *philip.torr@eng.ox.ac.uk*
*Department of Engineering Science, University of Oxford*

**Puneet K. Dokania**                                                           *puneet@robots.ox.ac.uk*
*Department of Engineering Science, University of Oxford, Five AI*

**Reviewed on OpenReview:** *https: // openreview. net/ forum? id= kfhoeZCeW7*

## Abstract

Pre-trained foundation models, due to their enormous capacity and exposure to vast amounts of data during pre-training, are known to have learned plenty of real-world concepts. An important step in making these pre-trained models effective on downstream tasks is to fine-tune them on related datasets. While various fine-tuning methods have been devised and have been shown to be highly effective, we observe that a fine-tuned model's ability to recognize concepts on tasks *different* from the downstream one is reduced significantly compared to its pre-trained counterpart. This is an undesirable effect of fine-tuning as a substantial amount of resources was used to learn these pre-trained concepts in the first place. We call this phenomenon "concept forgetting" and via experiments show that most end-to-end fine-tuning approaches suffer heavily from this side effect. To this end, we propose a simple fix to this problem by designing a new fine-tuning method called LDIFS (short for $\ell_2$ distance in feature space) that, while learning new concepts related to the downstream task, allows a model to preserve its pre-trained knowledge as well. Through extensive experiments on 10 fine-tuning tasks we show that LDIFS significantly reduces concept forgetting. Additionally, we show that LDIFS is highly effective in performing continual fine-tuning on a sequence of tasks as well, in comparison with both fine-tuning as well as continual learning baselines.

## 1 Introduction

Foundation models like CLIP Radford et al. (2021), ALIGN Jia et al. (2021) and CoCa Yu et al. (2022) are trained using self-supervised methods on hundreds of millions or even billions of samples scraped from the internet. This massive, compute intensive pre-training makes such models a knowledge store on a vast number of real-world concepts, enabling them to easily transfer to a wide variety of downstream tasks and applications. Indeed, this ability to recognize real-world concepts and thereby transfer to downstream tasks is the primary advantage of such models and is the very reason behind their name Bommasani et al. (2021).

While a foundation model can achieve impressive performance on a downstream task often without even requiring a single training sample from the task itself Radford et al. (2021), in order to maximise performance, it conventionally requires some form of fine-tuning on the task at hand. There are multiple types of fine-tuning methods like linear probing Radford et al. (2021), prompt-tuning Zhou et al. (2022b;a), adapters Gao et al. (2021); Zhang et al. (2021), weight-space interpolation Wortsman et al. (2022b); Ilharco et al. (2022a) and full

end-to-end fine-tuning Radford et al. (2021); Kumar et al. (2022); Goyal et al. (2022); Xuhong et al. (2018). Among these types, end-to-end fine-tuning is well-known to produce the best downstream performance.

It is worth noting that the pre-training dataset of a foundation model, owing to its massive scale, contains information about several thousands of real-world concepts. Hence, it is highly likely that the downstream dataset for fine-tuning the model will only contain a significantly smaller number of concepts compared to its pre-training set. A natural question that arises then is: *How does end-to-end fine-tuning of a foundation model affect the vast knowledge it acquired through its pre-training?* This is precisely what we aim to answer.

Through a thorough study of popular end-to-end fine-tuning methods, we observe that for most of them, the fine-tuned model has significantly lost its ability to recognize real-world concepts outside the downstream task, a phenomenon which we call *concept forgetting.* This is highly undesirable as there are important use-cases where we would want the foundation model's vast knowledge on concepts to be preserved even after fine-tuning.

One important use-case, for instance, is the requirement for *continual fine-tuning* to update a pre-trained model with previously unknown knowledge. Several organizations have spent millions of dollars pre-training large foundation models from scratch Knight (2023). Even after such powerful pre-training, these models can exhibit gaps in their knowledge. Specific examples include ChatGPT OpenAI (2020) having a knowledge-cutoff date of September 2021 or CLIP Radford et al. (2021) failing to perform in niche downstream areas like satellite or medical imagery. Furthermore, fields like medicine are constantly updating with new information like new diseases, new diagnoses and treatments etc. Therefore, having a fixed knowledge-base in medical foundation models like MedCLIP Wang et al. (2022a) or MedPaLM Singhal et al. (2022) is not an option if we are to use them in practical downstream use cases. With their expensive pre-training process, re-training a foundation model from scratch by combining new data with prior training data, is too expensive to be feasible. This requires the pre-trained foundation model to be fine-tuned on new data while preserving its prior knowledge, thereby necessitating investigation into fine-tuning methods which prevent concept forgetting.

Although conventional end-to-end fine-tuning methods generally suffer from concept forgetting, we find that there can be a relatively simple fix to this problem. In particular, if a fine-tuning method ensures that the fine-tuned model is close in some sense to the original foundation model, it can significantly reduce concept forgetting. One way to define the vicinity of the original model is in terms of distance in the parameter space. This is seen in the case of the L2SP regularizer Xuhong et al. (2018). However, we find that it is much more effective to define vicinity in terms of distance in the model's feature space which captures its input-output behaviour. This leads us to propose a new regularizer, **LDIFS ($\ell_2$ distance in Feature Space)**, which minimizes the distance between the features of the original model and the model being fine-tuned during fine-tuning. Furthermore, we observe that simply preserving the last-layer features is not effective in reducing concept forgetting. Motivated from observations in Zhang et al. (2018), we therefore preserve features extracted from different internal representations. We find that this relatively simple method of preserving the original model's features while fine-tuning on a downstream task can significantly alleviate the problem of concept forgetting in the fine-tuned model, without affecting its performance on the downstream task. Empirically we show this through an extensive evaluation of LDIFS on 10 different downstream tasks.

Finally, since LDIFS preserves pre-trained knowledge during fine-tuning, as a natural extension, we study a continual setup with a sequence of fine-tuning tasks. In particular, our continual setup uses 3 sequences, each of 3 tasks. Again, in every evaluation setting, we find LDIFS to outperform both fine-tuning as well as classic continual learning methods in minimizing concept forgetting, without compromising performance on the fine-tuned tasks themselves. Thus, to summarize, our contributions in this work are as follows:

1. **Investigate concept forgetting.** To the best of our knowledge, we are the first to perform a thorough analysis and evaluation of concept forgetting for fine-tuning multi-modal foundation models. We propose a simple way to quantify concept forgetting and benchmark 6 existing end-to-end fine-tuning methods on 10 different downstream tasks to find that concept forgetting, as a phenomenon, exists in all of them.

2. **Analyze different end-to-end fine-tuning methods.** We find a consistent ordering in concept forgetting for different fine-tuning methods. Particularly, we find the L2SP Xuhong et al. (2018) regularizer to outperform other fine-tuning baselines. We analyze why this is the case.

3. **Propose a new regularizer for end-to-end fine-tuning.** Analyzing L2SP helps us propose a simple new regularizer (LDIFS) which minimizes feature space distance between pre-trained and fine-tuned models during fine-tuning. Minimizing feature space distance fine-tunes the model on the downstream task while preserving much of the input-output behaviour of the pre-trained model. This helps LDIFS outperform other existing fine-tuning methods in minimizing concept forgetting during fine-tuning.

4. **A nudge towards continual fine-tuning.** Finally, as a natural extension of the evaluation setting we evaluate on a continual setup of 3 different sequences of 3 fine-tuning tasks each and find LDIFS to be superior to both fine-tuning methods as well as 5 classic continual learning baselines in preserving and accumulating knowledge in this setup.

## 2 A brief note on fine-tuning

Here we provide a description of CLIP Radford et al. (2021) as our foundation model of choice and briefly discuss existing state-of-the-art methods used for fine-tuning CLIP.

**CLIP, a brief overview** Broadly speaking, a CLIP model has two components: **i)** the vision or image encoder[1] $f_{\theta_v} : \mathbb{R}^{C,H,W} \to \mathbb{R}^D$, and **ii)** the text encoder $f_{\theta_t} : \mathbb{R}^L \to \mathbb{R}^D$. CLIP is pre-trained on 400 million pairs of images and corresponding text descriptions scraped from the internet. For pre-training, it uses a contrastive loss to maximize cosine similarity between the correct (image, text) pairs and minimize the same for the incorrect ones. Due to its large-scale self-supervised pre-training, CLIP exhibits impressive performance on several downstream tasks, often without requiring a single training sample from the task itself. However, in order to maximise performance on a specific downstream task, the pre-trained CLIP model is conventionally fine-tuned further on the task itself. Below we provide a brief description of such popular fine-tuning methods relevant to our study.

**Zero-shot (ZS)** In image classification, given an input image $\mathbf{x}$ and a set of $K$ class names $\{\mathbf{c}_i\}_{i=1}^K$ as natural language text, the $D$-dimensional encoding[2] for each class name $\psi(\mathbf{c}_i) = f_{\theta_t}(\mathbf{c}_i)$ and the image $\phi(\mathbf{x}) = f_{\theta_v}(\mathbf{x})$ are first obtained. The text encodings $\psi(\mathbf{c}_i)$ are then used as parameters of a $K$-class linear classifier, and the classification inference on $\mathbf{x}$ is performed as $\arg\max_i \psi(\mathbf{c}_i)^{\mathrm{T}} \phi(\mathbf{x})$. This is known as the zero-shot (ZS) prediction and CLIP's best model has been shown to have competitive ZS accuracies with a fully supervised ResNet-101 on ImageNet Radford et al. (2021).

**Linear Probe (LP)** In this case, an additional linear layer $\mathbf{w} \in \mathbb{R}^{D \times K}$ is appended on top of the image encoder $f_{\theta_v}$ and the weights of this linear layer are trained by solving a standard logistic regression problem (e.g., scikit-learn's `LogisticRegression` module Pedregosa et al. (2011)). The linear layer $\mathbf{w}$ is normally initialized using the text representations $\{\psi(\mathbf{c}_i)\}_{i=1}^K$, known as ZS initialization. It is trivial to note that in the absence of any training, LP boils down to ZS.

**End-to-end fine-tuning** While a pre-trained CLIP encoder can obtain impressive ZS and LP accuracies on several tasks, in order to maximize performance on a specific downstream task, the general rule of thumb is to initialize a model from the weights of the pre-trained encoder and then fine-tune the model end-to-end on the downstream task. Here we list some of the most popular end-to-end fine-tuning methods which we study in this work. We provide a more detailed discussion on the different types of fine-tuning methods in §6.

1. **ZS-init-CE** Radford et al. (2021): This is the classic end-to-end fine-tuning method where, similar to the LP, a ZS initialized linear head $\mathbf{w} : \mathbb{R}^D \to \mathbb{R}^K$ is appended to the image encoder $f_{\theta_v}$. However, differently from the LP, parameters of the entire model $\theta = \{\theta_v, \mathbf{w}\}$ (including the image encoder parameters) are fine-tuned using a cross-entropy loss $\mathcal{L}_{\mathrm{CE}}$.

2. **LP-init-CE (LP-FT)** Kumar et al. (2022): This is similar to ZS-init-CE but instead of initializing the appended linear head via ZS, it is initialized by performing linear probing on the downstream task first. Once the linear head is initialized, the entire model is end-to-end fine-tuned using $\mathcal{L}_{\mathrm{CE}}$.

3. **ZS-init-L2SP** Xuhong et al. (2018): In addition to the cross-entropy loss $\mathcal{L}_{\mathrm{CE}}$, this method uses an additional regularizer to minimize the $\ell_2$ distance between the pre-trained and fine-tuned *image encoder*

---

[1]Note, the vision encoder first extracts $D_v$ dimensional image features and then projects them to a $D$ dimensional space via a linear embedder $\mathbf{w}_v : \mathbb{R}^{D_v} \to \mathbb{R}^D$. Similarly for the text encoder.

[2]The natural language class names are often augmented with a prompt like "an image of a {class name}."

*weights*, thereby trying to keep the fine-tuned model weights close to the pre-trained ones. Let the pre-trained image encoder weights be $\theta_{v(0)}$ and the encoder weights at time step $t$ during fine-tuning be $\theta_{v(t)}$. Then, the fine-tuning loss in this case becomes

$$\mathcal{L}_{\text{L2SP}} = \mathcal{L}_{\text{CE}} + \lambda_{\text{L2SP}}||\theta_{v(t)} - \theta_{v(0)}||_2^2. \tag{1}$$

Note that the $\ell_2$ distance is only computed between the weights of the pre-trained and fine-tuned image encoders.

4. **LP-init-L2SP**: This is similar to ZS-init-L2SP but the linear head **w** is initialized by performing linear probing on the downstream dataset first. The loss for end-to-end fine-tuning then is the same as in Equation (1) [3].

5. **FLYP** Goyal et al. (2022): The Fine-tune Like You Pre-train or FLYP baseline fine-tunes both the image and the text encoders of CLIP and uses contrastive loss $\mathcal{L}_{\text{cont}}$ instead of cross-entropy $\mathcal{L}_{\text{CE}}$ for fine-tuning on the downstream task. The parameters being fine-tuned here are $\theta = \{\theta_v, \theta_t\}$, i.e., both image and text encoders of CLIP.

6. **FLYP-CE** Goyal et al. (2022): This is an ablation on FLYP where, instead of using contrastive loss, the fine-tuning is done using cross-entropy loss $\mathcal{L}_{\text{CE}}$, taking the cosine similarities between image and text embeddings as logits. Note that similar to FLYP, in this case as well, both image and text encoders are fine-tuned end-to-end.

## 3   The crippling effect of end-to-end fine-tuning

The contrastive pre-training dataset of CLIP contains 400 million (image, text) pairs scraped from the internet Radford et al. (2021). Consequently, any downstream task that CLIP is fine-tuned on is highly likely to contain only a small fraction of concepts compared to what it has already been exposed to during pre-training. To investigate the impact of fine-tuning, here we perform a thorough study benchmarking 6 fine-tuning methods on 10 downstream classification tasks. We find that for most of these methods, while the fine-tuned model attains excellent improved performance on the downstream task itself, its general ability to recognize concepts outside the task is significantly reduced over the course of fine-tuning. We call this phenomenon ***concept forgetting*** and find this to be an undesirable effect of most fine-tuning methods. To explore this in detail, in this section, we first discuss how we quantify concept forgetting, then we propose our benchmarking setup for fine-tuning methods and finally present our observations.

### 3.1   Quantifying Concept Forgetting

During ZS and LP evaluation of a model (refer §2) the pre-trained image encoder weights $\theta_v$ remain frozen and unchanged irrespective of the downstream task at hand. Therefore, ZS and LP performance on a specific downstream task can be a good indicator of the pre-trained model's existing knowledge about the task. This is the hypothesis we base our analysis on.

While ZS accuracy is based on how well the text encoder representations can form a linear classifier in the image encoder's feature space, LP performance is indicative of whether the image encoder representations are linearly separable in the first place. Note that this distinction is important. Fine-tuning the weights of just the image encoder $\theta_v$ can lead to a situation where its representations are no longer well-aligned with the text encoder. Even so, for a given task, if the image encoder representations are linearly separable, as captured by its LP accuracy, it shows that the model is still able to recognize concepts involved in the downstream task, thereby indicating the preservation of knowledge on the task. Thus, after fine-tuning, ZS accuracy may not reflect a model's ability to recognize concepts in a task and LP accuracy is a better candidate to do so. We illustrate this point in Figure 1.

Therefore, in order to quantify concept forgetting on a particular task defined on a dataset $\mathcal{D}$, we measure the difference in LP accuracy between the pre-trained and the fine-tuned image encoders on $\mathcal{D}$. To formalize

---

[3]To the best of our knowledge, LP-init-L2SP, has not been evaluated or benchmarked prior to this work.

this, let $f_{\theta_{v(0)}}$ be the pre-trained image encoder and $f_{\theta_v}$ the one obtained via fine-tuning on a dataset $\mathcal{D}_{\text{ft}}$. Furthermore, let $\mathcal{A}_{\text{LP}}(f_{\theta_v}, \mathcal{D})$ represent the LP accuracy of image encoder $f_{\theta_v}$ on dataset $\mathcal{D}$. Then, we define the change in LP accuracy on $\mathcal{D}$ between pre-trained and fine-tuned models $\Delta_{\text{LP}}(\mathcal{D}, f_{\theta_v}, f_{\theta_{v(0)}})$ (or in short, $\Delta_{\text{LP}}$) as:

$$\Delta_{\text{LP}}(\mathcal{D}, f_{\theta_v}, f_{\theta_{v(0)}}) = \mathcal{A}_{\text{LP}}(f_{\theta_v}, \mathcal{D}) - \mathcal{A}_{\text{LP}}(f_{\theta_{v(0)}}, \mathcal{D}) \tag{2}$$

Note, $f_{\theta_v}$ is fine-tuned on $\mathcal{D}_{\text{ft}}$, not on $\mathcal{D}$. The dataset $\mathcal{D}$ here represents a target dataset on which we would like to monitor/quantify concept forgetting. Clearly, if we use $\mathcal{D}_{\text{ft}}$ instead, we expect $\Delta_{\text{LP}}(\mathcal{D}_{\text{ft}}, f_{\theta_v}, f_{\theta_{v(0)}})$ to increase over the course of fine-tuning. However, the main objective here is to quantify the effect of fine-tuning on a task defined on a dataset $\mathcal{D}$ that was not part of the fine-tuning procedure itself ($\mathcal{D} \neq \mathcal{D}_{\text{ft}}$). A negative value of $\Delta_{\text{LP}}$ indicates **concept forgetting**, a zero indicates **knowledge accumulation** and a positive value indicates **knowledge gain** or positive forward transfer Lopez-Paz & Ranzato (2017) on the task under inspection. We would like to highlight that the dataset $\mathcal{D}$ here is a user-defined dataset on which monitoring the effect of fine-tuning would be desirable.

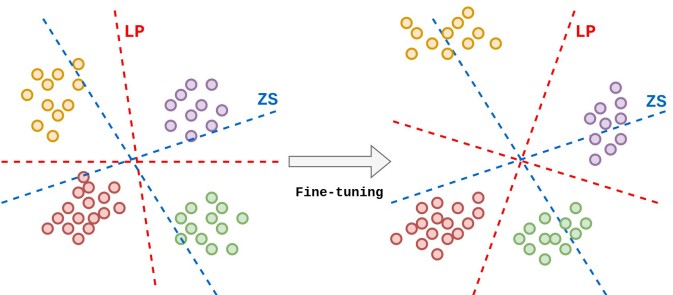

Figure 1: **Pictorial representation of classifiers from zero-shot (ZS) and linear probe (LP)**. While ZS can separate representations pre fine-tuning, the representations learnt after fine-tuning may not necessarily be linearly separable by the same ZS classifier anymore. However, if an LP is able to classify representations after fine-tuning, the model can still recognize the concepts involved.

**Catastrophic forgetting vs concept forgetting** Catastrophic forgetting McCloskey & Cohen (1989); Kemker et al. (2018); Kirkpatrick et al. (2017) is a well-known phenomenon which signifies how, when a model is trained on a new task, its performance on the previous task drops catastrophically. For example, if a model trained on ImageNet is then trained on say CIFAR100, it significantly loses its performance on ImageNet. Though *concept forgetting* mentioned in this work is very similar to catastrophic forgetting and we do not claim much conceptual novelty here, there is a subtle difference that we believe requires a distinction between the two. The major difference lies in the fact that in the pre-foundation model era, the pre-training datasets were much smaller and the pre-training tasks were fully supervised, for example, classification on the ImageNet dataset. Therefore, it was possible to roughly quantify the degree of damage the model had on the pre-trained task (using pre-trained dataset) once it was fine-tuned on a new downstream task. However, in the case of foundation models, it is not trivial to quantify what the model knows as it was pre-trained on several millions or even billions of examples using self-supervised training, and the pre-training dataset is also often inaccessible. Therefore, it is not possible to quantify exactly what the fine-tuned model forgot and the catastrophe therein. Hence, it is necessary to devise poking and probing mechanisms, similar to the ones we present in this work using LP, to quantify the effect of fine-tuning on a smaller but relevant domain of concepts (represented by the dataset $\mathcal{D}$ above).

## 3.2 Benchmarking concept forgetting

To quantify concept forgetting, here we use CLIP Radford et al. (2021) ViT-B/32 pre-trained on the OpenAI dataset and released in the OpenCLIP repository Ilharco et al. (2021) and measure its LP performance over fine-tuning on 10 different image classification downstream tasks with a high variability in their semantic concepts. These datasets, along with their respective train/test splits are:

1. *Stanford Cars* Krause et al. (2013) containing $16{,}185$ images of 196 classes of cars with a train/test split of 8144 and 8041 images respectively,

2. *CIFAR-10/100 (C10/100)* Krizhevsky et al. (2009) containing $60{,}000$ images of vehicles, flora and fauna, divided into 10/100 classes with the train/test split having $50{,}000$ and $10{,}000$ images respectively,

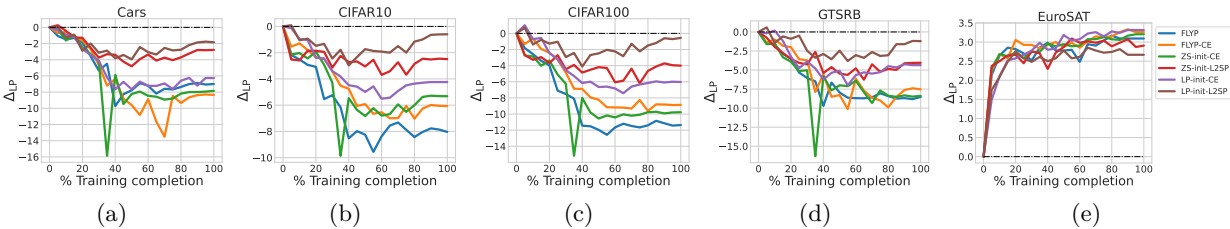

Figure 2: **Test set $\Delta_{\mathrm{LP}}$ for models fine-tuned on EuroSAT using different fine-tuning methods.** While EuroSAT $\Delta_{\mathrm{LP}}$ rises, $\Delta_{\mathrm{LP}}$ on all other datasets is almost always negative throughout the fine-tuning with a sole exception of DTD when fine-tuned using LP-init-L2SP. See §3.2 for details.

Figure 3: **Test set $\Delta_{\mathrm{LP}}$ evaluated on different datasets for models fine-tuned on EuroSAT via various fine-tuning methods.** The L2SP baselines (particularly LP-init-L2SP) have the lowest negative $\Delta_{\mathrm{LP}}$ on datasets other than EuroSAT.

3. *DTD* Cimpoi et al. (2014) containing 3760 images of 47 classes of textures found in the wild with 1880 images each in the train and test sets,

4. *EuroSAT* Helber et al. (2019) containing 25,000 samples with 10 categories of satellite images of landscapes and 19,600/5400 training/test images respectively,

5. *GTSRB* Stallkamp et al. (2012) containing 39,270 images of 43 classes of German traffic signs with 26,640 training images and 12,630 test images,

6. *MNIST* LeCun et al. (1998) containing 60,000 training images and 10,000 test images of 10 handwritten digits from 0 to 9 in grayscale,

7. *RESISC45* (R45) Cheng et al. (2017) containing 25,200 samples with 45 classes of various remote sensing image scenes with the train/test split having 18,900 and 6300 images respectively,

8. *SVHN* Netzer et al. (2011) containing a total of 99,289 colour images of street view house numbers, each image being categorized into one of 10 digits with 73,257 training samples and 26,032 test samples.

9. *ImageNet* Deng et al. (2009) containing a total of 1.28 million training images and 50,000 validation images of 1000 classes.

Finally, for this study, we use the 6 end-to-end fine-tuning methods discussed in §2. The training details can be found in Appendix A.

**Fine-tuning causes concept forgetting.** In Figure 2, we present the $\Delta_{\mathrm{LP}}$ for models fine-tuned on EuroSAT using the 6 fine-tuning methods discussed in §2. Across all fine-tuning methods, we observe that *while as expected, the performance on EuroSAT test increases and $\Delta_{\mathrm{LP}}$ is positive (also see Figure 3e), performance on 8 other downstream datasets decreases and $\Delta_{\mathrm{LP}}$ for all of them is negative.* This indicates that *all 6 fine-tuning methods suffer from concept forgetting.* The only exception to this is when we evaluate on DTD and the model is fine-tuned using LP-init-L2SP. We observe $\Delta_{\mathrm{LP}}$ to be mostly positive before it goes to zero at the end of fine-tuning. This is an interesting case as it indicates that LP-init-L2SP on EuroSAT might actually be helping increase knowledge about DTD before the fine-tuning becomes too specific to EuroSAT. This may be an example of positive forward transfer Lopez-Paz & Ranzato (2017); Chaudhry et al. (2018) and exploring why such knowledge accumulation happens is an interesting avenue for further research.

Next, we compare between these fine-tuning methods by showing $\Delta_{\mathrm{LP}}$ for different downstream datasets in Figure 3, where it is evident that *LP-init-L2SP consistently outperforms other baselines in lowering concept forgetting and preserving the fine-tuned model's original performance across multiple downstream tasks.* This is observable from its low negative $\Delta_{\mathrm{LP}}$ compared to other baselines. While it suffers an initial dip in

Figure 4: $\ell_2$ **distance in the parameter space** $||\theta_{v(t)} - \theta_{v(0)}||_2^2$ of the image encoder over the course of fine-tuning. Except L2SP, all other baselines diverge away from their pre-trained counterparts.

performance in the early stages of fine-tuning, in the later stages, LP-init-L2SP regains the accuracy, often ending up with a near zero $\Delta_{LP}$. Its impressive performance on concept forgetting however, does seem to come at the cost of a relatively lower $\Delta_{LP}$ on the fine-tuning task, i.e., EuroSAT, itself. In this regard, note that most fine-tuning methods are specifically designed to maximise performance on the downstream fine-tuning task. Thus, the performance in Figure 3e is conventionally the primary evaluation criterion for a fine-tuning method. However, as shown here, concept forgetting is an undesirable additional effect of fine-tuning and requires its own evaluation. Our observations for other datasets are similar (see Appendix C). In what follows, we first investigate LP-init-L2SP further to gain insights on why it preserves concepts better than other baselines, and then use those insights to propose a new fine-tuning method that significantly outperforms all other baselines.

## 4 Can preserving features help?

The L2SP regularizer (eq. (1)) enforces the model $f_{\theta_{v(t)}}$ at time-step $t$ of fine-tuning to be in the vicinity of the pre-trained model $f_{\theta_{v(0)}}$ by minimizing the $\ell_2$ distance between the two in the **parameter space**. As evident from §3, this simple regularizer has a promising impact on the model's ability to avoid concept forgetting. To understand how correlated the parameter space $\ell_2$ distance $||\theta_{v(t)} - \theta_{v(0)}||_2^2$ is to concept forgetting, in Figure 4, we show how $\ell_2$ distance changes as we fine-tune different datasets (EuroSAT, GTSRB and SVHN) using all the discussed fine-tuning methods. *From Figure 4, relatively speaking, all the fine-tuning methods except the two L2SP baselines (ZS-init-L2SP and LP-init-L2SP) cause the parameters of the fine-tuned model to move away from its pre-trained counterpart.*

Though regularizing the fine-tuned model to be in the vicinity of the pre-trained one shows a promising effect in preserving concepts, vicinity in the parameter space need not necessarily capture the input-output behaviour of the pre-trained model. In fact, it is trivial to construct two different sets of model parameters, far in $\ell_2$ distance, outputting similar values in a specific domain. Additionally, regularizing in the parameter space might not keep the fine-tuned model in the *desired* vicinity that preserves the pre-trained knowledge and performs well on the downstream task at the same time. Therefore, we conjecture that the vicinal distance in feature space as opposed to parameter space, might be a better indicator of the similarity of encoded concepts in the model. Indeed, in the end, the internal representation space is where models encode patterns. Thus, in Figure 5, we inspect $\ell_2$ distance in the feature space over fine-tuning using the following distance function:

$$d(\theta_{v(t)}, \theta_{v(0)}, \mathcal{D}) = \frac{1}{N} \sum_{i=1}^{N} ||\Phi_{\theta_{v(t)}}(\mathbf{x}_i) - \Phi_{\theta_{v(0)}}(\mathbf{x}_i)||_2^2, \tag{3}$$

where, $\Phi_{\theta_{v(t)}}(\mathbf{x}_i)$ represents the features of the model with parameters $\theta_{v(t)}$ at time $t$ for a given sample $\mathbf{x}_i$, and $N$ is the number of samples in the dataset. Note that the feature vector $\Phi_{\theta_{v(t)}}(\mathbf{x}_i)$ is obtained by concatenating various internal representations (not just the last layer features) of the network architecture, similar to the perceptual features presented in Zhang et al. (2018). The exact details as to how we compute the concatenated feature vector for a ViT-B/32 model is mentioned in Appendix B, and similar plots for other datasets is shown in Appendix C.

**A strong correlation between concept forgetting and feature-space distance.** Similar to our observations in Figure 4 (parameter space), we note that except L2SP, all other fine-tuning methods suffering significantly from concept forgetting cause the fine-tuned model to move away from the pre-trained model

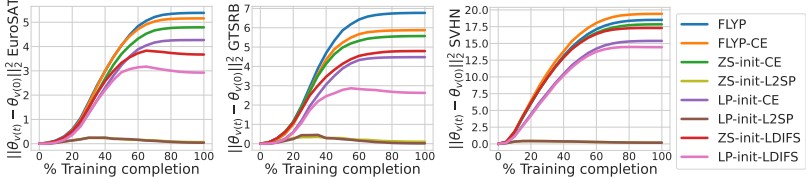

Figure 5: $\ell_2$ **distance in feature space** $d(\theta_{v(t)}, \theta_{v(0)}, \mathcal{D})$ (see eq. (3)) between image encoders computed over the course of fine-tuning on EuroSAT. The $\ell_2$ distance is computed for EuroSAT train and test sets as well as CIFAR10, CIFAR100 and GTSRB datasets.

Figure 6: $\ell_2$ **distance in weight space** $||\theta_{v(t)} - \theta_{v(0)}||_2^2$ between the image encoder fine-tuned to the current time-step $f_{\theta_{v(t)}}$ and the pre-trained image encoder $f_{\theta_{v(0)}}$ over the course of fine-tuning.

in terms of feature-space distance as well (Equation (3)). In case of ZS-init-L2SP and LP-init-L2SP, while initially diverging, the fine-tuned models recover their pre-trained behaviour to a certain extent in the later stages of fine-tuning. It is important to note that this observation is consistent for the fine-tuning training and test sets as well as for all other datasets.

Based on the results shown in Figure 3 and Figure 5 we conclude that the *fine-tuning methods which cause the model to significantly diverge away from the pre-trained model either in parameter space or in feature space suffer more from concept forgetting*. This observation, combined with our conjecture above leads to a natural extension of fine-tuning where we use $d(\theta_{v(t)}, \theta_{v(0)}, \mathcal{D})$ (distance in the feature space) as the regularizer. We call this regularizer **LDIFS: $\ell_2$ distance in Feature Space**. The complete fine-tuning objective then becomes:

$$\mathcal{L}_{\text{LDIFS}} = \mathcal{L}_{\text{CE}} + \lambda_{\text{LDIFS}} \cdot d(\theta_{v(t)}, \theta_{v(0)}, \mathcal{D}_{\text{train}}) \tag{4}$$

where $\mathcal{D}_{\text{train}}$ is the training or fine-tuning set and $\lambda_{\text{LDIFS}}$ is the regularization coefficient. Note that similar to L2SP, we can initialize the linear head with both zero-shot weights or linear probe weights leading to two variants: ZS-init-LDIFS and LP-init-LDIFS.

**Analysing LDIFS on parameter space and feature space distance:** In Figure 6 and Figure 7, we plot the $\ell_2$ distance in parameter/weight and feature space respectively (same as Figure 4 and Figure 5), but with the LDIFS added in. In the weight space, LDIFS, while lower than other fine-tuning methods, has a relatively higher $\ell_2$ distance from the pre-trained model compared to L2SP. On the contrary, in the feature space, LDIFS consistently gets the lowest distance from the pre-trained model. This is expected and was indeed the purpose behind LDIFS's design which is to minimize the difference in input-output behaviour, captured through feature space distance between the fine-tuned and pre-trained models and not necessarily constrain the parameter space of the fine-tuned model to lie close to the pre-trained model. Finally, note from Figure 7, that although we are applying the LDIFS regularizer only on the EuroSAT samples during fine-tuning, the preservation of features extends even to other datasets like CIFAR10, CIFAR100 and GTSRB, as seen from the low feature space distance even on these datasets. This indicates, that we don't need third party datasets during fine-tuning to preserve the pre-trained model's behaviour and regularizing on samples just from the fine-tuning dataset can be sufficient.

**LP-init-LDIFS significantly reduces concept forgetting.** First, we evaluate LDIFS on the same setting as in §3 and present the $\Delta_{\text{LP}}$ over fine-tuning in Figure 8 (additional results presented in Appendix C). Second, in Table 1, we report the $\mathcal{A}_{\text{LP}}\%$ on the fine-tuning test set and the $\Delta_{\text{LP}}$ averaged over other tasks. $\mathcal{A}_{\text{LP}}\%$ measures performance on the fine-tuning task itself and $\Delta_{\text{LP}}$ provides an estimate of the fine-tuned model's level of concept forgetting on the remaining tasks. For downstream tasks which are not ImageNet, we report $\Delta_{\text{LP}}$ averaged over the remaining 8 tasks, except ImageNet. For ImageNet, we leave out CIFAR-10/100 when calculating performance over other tasks since all of CIFAR-10/100's classes are within the set of

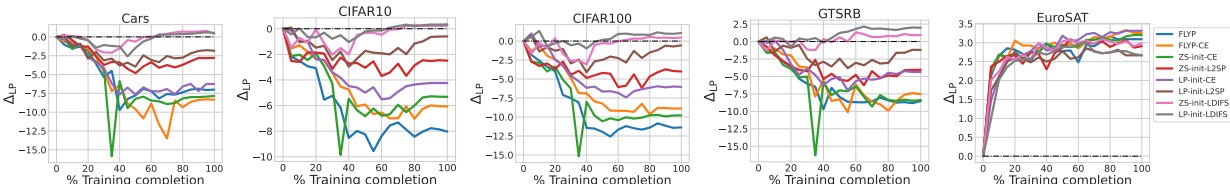

Figure 7: $\ell_2$ **distance in feature space** $d(\theta_{v(t)}, \theta_{v(0)}, \mathcal{D})$, between image encoders, $f_{\theta_{v(t)}}$ and $f_{\theta_{v(0)}}$ computed over different fine-tuning methods for models fine-tuned on EuroSAT.

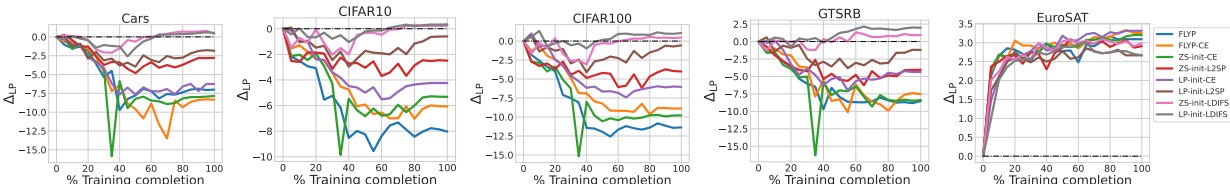

Figure 8: **Test set $\Delta_{\mathrm{LP}}$ evaluated on different datasets for models fine-tuned on EuroSAT.** LDIFS (Our) baselines (both ZS-init and LP-init) provides the best results in terms of avoiding concept forgetting without affecting downstream performance on EuroSAT.

| Dataset | Fine-tuning baselines | | | | | | | | | | | | | | | |
| | **FLYP** | | **FLYP-CE** | | **ZS-init-CE** | | **LP-init-CE** | | **ZS-init-L2SP** | | **LP-init-L2SP** | | **ZS-init-LDIFS** | | **LP-init-LDIFS** | |
| | $\mathcal{A}_{\mathrm{LP}}(\uparrow)$ | $\Delta_{\mathrm{LP}}(\uparrow)$ | $\mathcal{A}_{\mathrm{LP}}(\uparrow)$ | $\Delta_{\mathrm{LP}}(\uparrow)$ | $\mathcal{A}_{\mathrm{LP}}(\uparrow)$ | $\Delta_{\mathrm{LP}}(\uparrow)$ | $\mathcal{A}_{\mathrm{LP}}(\uparrow)$ | $\Delta_{\mathrm{LP}}(\uparrow)$ | $\mathcal{A}_{\mathrm{LP}}(\uparrow)$ | $\Delta_{\mathrm{LP}}(\uparrow)$ | $\mathcal{A}_{\mathrm{LP}}(\uparrow)$ | $\Delta_{\mathrm{LP}}(\uparrow)$ | $\mathcal{A}_{\mathrm{LP}}(\uparrow)$ | $\Delta_{\mathrm{LP}}(\uparrow)$ | $\mathcal{A}_{\mathrm{LP}}(\uparrow)$ | $\Delta_{\mathrm{LP}}(\uparrow)$ |
| --- | --- | --- | --- | --- | --- | --- | --- | --- | --- | --- | --- | --- | --- | --- | --- | --- |
| Cars | 86.06 | −0.01 | 84.77 | −1.67 | 83.48 | −1.56 | 84.95 | −0.63 | 82.64 | −1.07 | 83.87 | **0.47** | 85.52 | −0.36 | 85.26 | −0.18 |
| CIFAR10 | 97.71 | −3.35 | 97.58 | −1.17 | 97.73 | −1.6 | 97.71 | −0.81 | 97.7 | 1.04 | 97.66 | 1.16 | 97.36 | 1.03 | 97.24 | **1.18** |
| CIFAR100 | 88.98 | −1.16 | 88.77 | −0.5 | 88.6 | −0.96 | 88.41 | −0.11 | 87.84 | 0.65 | 86.94 | **1.03** | 87.94 | 0.82 | 88.99 | 0.86 |
| DTD | 76.12 | −4.92 | 73.46 | −3.44 | 77.18 | −3.01 | 72.18 | −1.76 | 72.98 | −3.71 | 74.63 | 0.01 | 78.14 | 0.19 | 75.27 | **0.53** |
| EuroSAT | 98.65 | −6.7 | 98.8 | −5.44 | 98.76 | −5.72 | 98.87 | −3.75 | 98.46 | −2.58 | 98.2 | −0.85 | 98.54 | 0.92 | 98.22 | **1.32** |
| GTSRB | 99.26 | −8.5 | 99.0 | −3.76 | 98.52 | −5.9 | 98.53 | −0.94 | 97.4 | −3.05 | 95.0 | 1.18 | 98.45 | 1.07 | 97.81 | **1.27** |
| MNIST | 99.62 | −8.64 | 99.63 | −7.53 | 99.67 | −8.76 | 99.68 | −6.02 | 99.43 | −2.93 | 99.18 | 1.49 | 99.6 | 1.8 | 99.52 | **2.64** |
| RESISC45 | 95.84 | −5.42 | 95.79 | −3.32 | 95.76 | −3.79 | 95.56 | −2.27 | 94.05 | −0.91 | 94.13 | 0.66 | 95.22 | 0.41 | 95.13 | **0.9** |
| SVHN | 97.44 | −10.74 | 97.4 | −10.40 | 97.3 | −11.12 | 97.5 | −8.73 | 96.5 | −2.78 | 96.54 | −2.11 | 96.97 | −0.68 | 96.95 | **−0.29** |
| ImageNet | 82.26 | −1.6 | 82.18 | −1.39 | 82.02 | −1.26 | 82.12 | −0.87 | 80.9 | −0.24 | 80.78 | −0.1 | 82.14 | 0.13 | 82.21 | **0.35** |

Table 1: **Test set accuracy $\mathcal{A}_{\mathrm{LP}}\%$ and average $\Delta_{\mathrm{LP}}$ computed over other datasets for models fine-tuned on 10 image classification tasks.** $\mathcal{A}_{\mathrm{LP}}$ shows performance on the fine-tuning task itself and $\Delta_{\mathrm{LP}}$ shows the level of concept forgetting on other tasks (higher $\Delta_{\mathrm{LP}}$ shows lower concept forgetting and vice versa). The best $\Delta_{\mathrm{LP}}$ numbers are shown in bold.

ImageNet classes. We use the remaining 7 tasks to quantify forgetting. Full set of results for all baselines can be found in Table 9 and an alternative visualization of these results are in Figure 14 of the Appendix. Our observations are as follows.

1. ***Competitive downstream accuracy***: From Table 1, we note that there is no clear winner in terms of accuracy on the downstream fine-tuning task itself. Additionally, from Figure 8 and Table 1, it is also evident that LDIFS is competitive with other fine-tuning baselines in terms of accuracy on the downstream fine-tuning task itself.

2. ***Minimal concept forgetting***: *LP-init-LDIFS, significantly minimizes concept forgetting over the course of fine-tuning.* This is evident from Figure 8 which shows its noticeably higher $\Delta_{\mathrm{LP}}$ on other downstream tasks over the course of fine-tuning. It is also apparent from its consistently high $\Delta_{\mathrm{LP}}$ scores in Table 1. Particularly, LP-init-LDIFS gets the highest average $\Delta_{\mathrm{LP}}$ on other tasks for 8 out of 10 fine-tuning cases. This indicates a significantly lower level of concept forgetting.

3. ***Positive Forward Transfer***: From Table 1, we also observe that in 8 out of 10 fine-tuning tasks, LP-init-LDIFS results in a positive average $\Delta_{\mathrm{LP}}$ on other tasks, thereby generally achieving a positive forward transfer. This is not the case for other baselines where $\Delta_{\mathrm{LP}}$ is generally negative indicating concept forgetting. The two exceptions for LP-init-LDIFS are when the finetuning tasks $\mathcal{D}_{\mathrm{ft}}$ are Stanford Cars and SVHN, in which case the average $\Delta_{\mathrm{LP}}$ on other tasks is negative. This observation also highlights how the ordering of fine-tuning tasks can also impact forward transfer. For instance, from Table 9, when fine-tuning on EuroSAT, LP-init-LDIFS achieves a $\Delta_{\mathrm{LP}}$ of +3.03 on SVHN, whereas in the reverse scenario, fine-tuning on SVHN leads to a negative $\Delta_{\mathrm{LP}}$ of −0.78 on EuroSAT.

| Model | Architecture | FLYP | | FLYP-CE | | ZS-init-CE | | LP-init-CE | | ZS-init-L2SP | | LP-init-L2SP | | ZS-init-LDIFS | | LP-init-LDIFS | |
|---|---|---|---|---|---|---|---|---|---|---|---|---|---|---|---|---|---|
| | | $\mathcal{A}_{LP}(\uparrow)$ | $\Delta_{LP}(\uparrow)$ | $\mathcal{A}_{LP}(\uparrow)$ | $\Delta_{LP}(\uparrow)$ | $\mathcal{A}_{LP}(\uparrow)$ | $\Delta_{LP}(\uparrow)$ | $\mathcal{A}_{LP}(\uparrow)$ | $\Delta_{LP}(\uparrow)$ | $\mathcal{A}_{LP}(\uparrow)$ | $\Delta_{LP}(\uparrow)$ | $\mathcal{A}_{LP}(\uparrow)$ | $\Delta_{LP}(\uparrow)$ | $\mathcal{A}_{LP}(\uparrow)$ | $\Delta_{LP}(\uparrow)$ | $\mathcal{A}_{LP}(\uparrow)$ | $\Delta_{LP}(\uparrow)$ |
| CLIP | RN50 | 78.64 | −4.18 | 78.42 | −3.92 | 78.39 | −4.01 | 78.45 | −3.4 | 76.2 | −2.1 | 76.13 | −1.54 | 77.94 | −0.67 | 78.16 | **−0.11** |
| | VB32 | 82.26 | −3.74 | 82.18 | −2.46 | 82.02 | −3.02 | 82.12 | −2.17 | 80.9 | −1.13 | 80.78 | −0.88 | 82.14 | 0.02 | 82.21 | **0.1** |
| | VB16 | 85.6 | −2.87 | 85.38 | −2.16 | 85.36 | −2.92 | 85.36 | −1.73 | 83.11 | −1.03 | 82.19 | −0.74 | 85.24 | 0.07 | 85.31 | **0.16** |
| | VL14 | 88.01 | −2.4 | 87.96 | −1.6 | 87.88 | −2.33 | 87.91 | −1.52 | 86.94 | −0.87 | 86.87 | −0.43 | 87.74 | 0.12 | 87.85 | **0.22** |
| FLAVA | VB16 | 81.79 | −4.07 | 81.54 | −3.72 | 81.18 | −3.94 | 81.36 | −3.04 | 79.67 | −1.82 | 80.11 | −1.1 | 81.24 | −0.34 | 81.61 | **0.04** |
| DINOv2 | VB14 | - | - | - | - | 85.32 | −2.71 | 85.48 | −1.86 | 84.16 | −0.92 | 84.5 | −0.66 | 85.63 | 0.01 | 86.02 | **0.06** |
| | VL14 | - | - | - | - | 87.6 | −1.92 | 87.9 | −1.4 | 86.63 | −0.64 | 87.02 | −0.19 | 87.84 | 0.08 | 87.91 | **0.13** |
| MAE | VB16 | - | - | - | - | 83.57 | −5.1 | 83.81 | −4.36 | 82.7 | −3.87 | 82.84 | −3.03 | 83.39 | −1.62 | 83.76 | **−0.94** |
| | VL16 | - | - | - | - | 85.86 | −4.26 | 86.04 | −3.59 | 85.02 | −2.64 | 85.1 | −1.82 | 85.47 | −0.75 | 85.9 | **−0.12** |

Table 2: **Test set accuracy $\mathcal{A}_{LP}$% and average $\Delta_{LP}$** computed for different architectures of CLIP Radford et al. (2021), FLAVA Singh et al. (2022), DINOv2 Oquab et al. (2023) and MAE He et al. (2022) all fine-tuned on ImageNet. The best $\Delta_{LP}$ numbers are shown in bold.

We have similar conclusions on 3 other CLIP architectures: ResNet-50, VIT-B/16 and ViT-L/14, a FLAVA ViT-B/16 Singh et al. (2022), a DINO-v2 ViT-B/14 and ViT-L/14 Oquab et al. (2023) and Masked Auto-encoder He et al. (2022) ViT-B/16 and ViT-L/16 models. We present the details and results next.

**Generality of concept forgetting and LDIFS beyond CLIP ViT-B/32.** To study the relevance of concept forgetting and LDIFS beyond a CLIP ViT-B/32 model, we experiment with other architectures and models. Among text-image multi-modal foundation models, we use CLIP and FLAVA Singh et al. (2022). For CLIP, we add the ResNet-50 (RN50), ViT-B/16 (VB16) and ViT-L/14 (VL14) architectures in addition to ViT-B/32. For FLAVA, we use the ViT-B/16 architectue. In addition, we add experiments on vision only foundation and pre-trained models as well, in particular DINOv2 Oquab et al. (2023) and Masked Auto-Encoders (MAE) He et al. (2022). For DINOv2, we use ViT-B/14 and ViT-L/14 architectures and for MAE, we use ViT-B/16 and ViT-L/16 architectures. All the models were fine-tuned on ImageNet using the same training recipe as mentioned in Appendix A. To quantify concept forgetting, we use DTD, EuroSAT, GTSRB, RESISC45 and SVHN as other datasets and report $\Delta_{LP}$ averaged over these 5 datasets. Note that the FLYP and FLYP-CE baselines particularly work with multi-modal foundation models, so we don't evaluate these two baselines for DINOv2 and MAE. Finally, ZS-init fine-tuning on DINOv2 and MAE are done using a randomly initialized linear head. We present our results from this experiment in Table 2. Following are our observations:

1. **Concept forgetting happens across architectures and foundation models.** From Table 2, we can see a consistently negative $\Delta_{LP}$ indicating concept forgetting for all models and architectures and all classic fine-tuning baselines (except our proposed LDIFS baseline).

2. **Larger models forget less.** For each model category, i.e., CLIP, FLAVA, DINOv2 and MAE, we observe progressively lower concept forgetting as model size gets larger. The ViT-L models consistently forget less than their ViT-B counterparts.

3. **CLIP and DINOv2 are competitve and outperform FLAVA and MAE.** However, it is not clear if this is an artefact of larger pre-training sets for CLIP (WIT-400M) and DINOv2 (LVD-142M) as opposed to FLAVA (PMD-70M) and MAE (ImageNet-1K 1.3M), their pre-training self-supervised loss functions or some combination of other factors. This is interesting to investigate for future work.

4. **LP-init-LDIFS is the consistent winner.** For each model and each architecture, we find our proposed LP-init-LDIFS to provide the highest $\Delta_{LP}$ values which are mostly positive, indicating both minimized concept forgetting as well as positive forward transfer in many cases.

## 5  A nudge towards continual fine-tuning

Our results above indicate that fine-tuning on LP-init-LDIFS can make the foundation model learn new downstream information without forgetting pre-trained concepts. A natural question that then arises is: *Can we fine-tune on a sequence of downstream tasks without forgetting concepts?* For an ideal fine-tuning method, the final model should attain state-of-the-art performance on all fine-tuned tasks while still maintaining its pre-trained knowledge. We empirically investigate this question by training on 3 sequences of 3 tasks each: **a)** SVHN → C10 → R45, **b)** SVHN → C100 → R45 and **c)** SVHN → Cars → R45. Note that this setup

Figure 9: $\mathcal{A}_{\text{LP}}$ for sequence: SVHN → CIFAR10 → RESISC45 (left 3 plots) and ii) SVHN → CIFAR100 → RESISC45 (right 3 plots). Vertical red line indicates a switch in the fine-tuning tasks.

is similar to continual learning Chaudhry et al. (2018); Rebuffi et al. (2017); Lopez-Paz & Ranzato (2017) but for pre-trained foundation models. Due to their impressive performance on a wide range of downstream tasks, continual fine-tuning of foundation models is relatively unexplored in the literature. Nonetheless, as stated in §1, this is an important problem to investigate from the perspective of updating a foundation model with new previously unknown knowledge without forgetting previously known ones.

**Quantifying concept forgetting in continual setups.** Let the sequence of fine-tuning datasets be $\mathcal{D}_1 \to \mathcal{D}_2, \to ..., \mathcal{D}_k$ and the corresponding sequence of models be $f_{\theta_0}, ..., f_{\theta_k}$ with $f_{\theta_0}$ being the pre-trained model. In order to then quantify concept forgetting on a task $\mathcal{D}$ over this sequence of models, we extend the $\Delta_{\text{LP}}$ from Equation (2) as follows:

$$\Delta_{\text{LP}}(\mathcal{D}, f_{\theta_k}, \{f_{\theta_0}, f_{\theta_1}, ..., f_{\theta_{k-1}}\}) = \mathcal{A}_{\text{LP}}(f_{\theta_k}, \mathcal{D}) - \max_{i \in \{0, ..., k-1\}} \mathcal{A}_{\text{LP}}(f_{\theta_i}, \mathcal{D}). \tag{5}$$

Hence, similar to Chaudhry et al. (2018), we find the difference in LP performance between the final fine-tuned model $f_{\theta_k}$ and the model having the maximum LP performance in the sequence $\{f_{\theta_0}, f_{\theta_1}, ..., f_{\theta_{k-1}}\}$. This accounts for the possibility of positive forward transfer on $\mathcal{D}$ over the sequence of fine-tuning tasks.

**LP-init-LDIFS significantly reduces concept forgetting in continual setups.** In Table 3, we present the $\Delta_{\text{LP}}$ and $\mathcal{A}_{\text{LP}}$ for the continual setup for all fine-tuning baselines. For each task sequence, we report the performance on datasets in the sequence in the first three rows and in the fourth row, we report the average performance on 6 other datasets. From Table 3, we observe:

1. For all 3 sequences, LP-init-LDIFS has minimal concept forgetting on tasks which appear earlier in the fine-tuning sequence, as well as other datasets which are not used for fine-tuning.
2. At the same time, LP-init-LDIFS is very competitive on accuracy with the best fine-tuning methods and consistently outperforms L2SP on accuracy on the last fine-tuning task, i.e., RESISC45.

These observations are further complemented in Figure 9 where we can see LP-init-LDIFS to have minimal forgetting on prior tasks, SVHN and C10/100 without compromising on performance on the last task.

**LP-init-LDIFS outperforms classic continual learning methods.** Due to the similarity of the proposed setup in this section with classic continual learning, we empirically compare performance of LP-init-LDIFS with 5 well-known continual learning baselines: LwF Li & Hoiem (2017), LFL Jung et al. (2016), iCARL Rebuffi et al. (2017), Distillation + Retrospection (D+R) Hou et al. (2018) and ZSCL Zheng et al. (2023) on our sequence setup. Results are in Table 4. Again, the results indicate that LP-init-LDIFS performs better than all other continual learning baselines, both in preventing forgetting as well as obtaining the best performance on the last task. This is evident from its consistently high $\Delta_{\text{LP}}$ and $\mathcal{A}_{\text{LP}}$ on all 3 sequences. Finally, in Appendix C.7, we provide additional results on an ablation comparing LP-init-LDIFS with joint training Li & Hoiem (2017) on our 3-task setting. In addition, we provide a comparison of LP-init-LDIFS with continual learning methods Wang et al. (2022c;b); Smith et al. (2023); Thengane et al. (2022); Zhang et al. (2023) which utilize pre-trained encoders to improve performance. We perform this comparison on Split ImageNet-R Wang et al. (2022b), a well-known class-incremental learning benchmark. In these experiments, LDIFS outperforms baselines which use pre-training and performs competitively with joint training.

# 6 Related Work & Discussion

In this section, we first provide a brief discussion on the different types of fine-tuning approaches apart from end-to-end fine-tuning which have been applied on foundation models. We also discuss the relation and differences between our proposed LDIFS regularizer and previous works on continual learning which particularly use feature-distillation methods.

| Dataset Fine-tune | Eval | FLYP | | FLYP-CE | | ZS-init-CE | | Fine-tuning baselines LP-init-CE | | ZS-init-L2SP | | LP-init-L2SP | | ZS-init-LDIFS | | LP-init-LDIFS | |
|---|---|---|---|---|---|---|---|---|---|---|---|---|---|---|---|---|---|
| | | $\Delta_{LP}(\uparrow)$ | $\mathcal{A}_{LP}(\uparrow)$ | $\Delta_{LP}(\uparrow)$ | $\mathcal{A}_{LP}(\uparrow)$ | $\Delta_{LP}(\uparrow)$ | $\mathcal{A}_{LP}(\uparrow)$ | $\Delta_{LP}(\uparrow)$ | $\mathcal{A}_{LP}(\uparrow)$ | $\Delta_{LP}(\uparrow)$ | $\mathcal{A}_{LP}(\uparrow)$ | $\Delta_{LP}(\uparrow)$ | $\mathcal{A}_{LP}(\uparrow)$ | $\Delta_{LP}(\uparrow)$ | $\mathcal{A}_{LP}(\uparrow)$ | $\Delta_{LP}(\uparrow)$ | $\mathcal{A}_{LP}(\uparrow)$ |
| SVHN → C10 → R45 | SVHN | −7.06 | 90.3 | −5.77 | 91.61 | −7.13 | 90.29 | −6.46 | 90.97 | −5.41 | 91.01 | −4.53 | 91.93 | −0.43 | 96.66 | −0.41 | 96.68 |
| | CIFAR10 | −3.16 | 94.61 | −1.92 | 95.65 | −2.31 | 95.25 | −1.57 | 96.33 | −1.22 | 96.33 | −0.25 | 97.26 | −0.26 | 97.18 | −0.21 | 97.41 |
| | RESISC45 | 4.06 | 95.33 | 3.89 | 95.16 | 4.0 | 95.3 | 2.98 | 94.29 | 2.97 | 94.24 | 2.16 | 93.44 | 3.94 | 95.33 | 3.7 | 95.0 |
| | Others | −6.59 | 78.91 | −4.9 | 81.2 | −5.08 | 80.91 | −4.24 | 82.13 | −1.82 | 85.27 | −0.01 | 86.89 | −0.3 | 86.52 | 0.1 | 87.08 |
| SVHN → C100 → R45 | SVHN | −7.75 | 89.64 | −6.71 | 90.65 | −7.28 | 90.05 | −2.73 | 94.42 | −11.36 | 85.18 | −6.12 | 90.42 | −1.5 | 95.47 | −0.65 | 96.32 |
| | CIFAR100 | −8.85 | 79.22 | −6.38 | 81.55 | −7.18 | 81.08 | −3.04 | 82.63 | −3.46 | 84.2 | −0.88 | 85.72 | −0.99 | 86.45 | −0.3 | 86.54 |
| | RESISC45 | 4.38 | 95.68 | 3.9 | 95.21 | 4.13 | 95.4 | 2.51 | 93.81 | 2.79 | 94.13 | 1.9 | 93.21 | 4.29 | 95.59 | 3.83 | 95.11 |
| | Others | −5.23 | 83.09 | −4.68 | 83.97 | −4.65 | 83.76 | −4.02 | 85.14 | −1.61 | 87.52 | −0.37 | 89.04 | −0.56 | 88.36 | −0.23 | 89.12 |
| SVHN → Cars → R45 | SVHN | −1.26 | 96.07 | −1.51 | 95.76 | −1.45 | 95.93 | −0.76 | 96.58 | −1.14 | 95.42 | −0.44 | 95.98 | −0.49 | 96.56 | −0.17 | 96.9 |
| | Cars | −3.61 | 81.21 | −4.3 | 76.87 | −4.18 | 76.96 | −8.36 | 71.6 | −0.88 | 81.18 | −0.4 | 81.82 | −0.61 | 82.89 | 0.47 | 84.23 |
| | RESISC45 | 4.13 | 95.43 | 3.81 | 95.08 | 3.89 | 95.17 | 3.0 | 94.35 | 2.83 | 94.13 | 2.13 | 93.43 | 3.94 | 95.22 | 3.73 | 95.27 |
| | Others | −6.68 | 81.91 | −5.48 | 83.2 | −4.93 | 83.38 | −4.51 | 84.39 | −2.91 | 85.74 | −1.67 | 87.15 | −0.1 | 88.75 | 0.23 | 89.39 |

Table 3: $\Delta_{LP}$ and $\mathcal{A}_{LP}\%$ for models fine-tuned on (SVHN, C10, R45), (SVHN, C100, R45) & (SVHN, Cars, R45) sequences. The first 3 rows show performance on fine-tuned tasks and the third row shows averaged performance on 6 other datasets.

| Dataset Fine-tune | Eval | LwF | | LFL | | Continual Learning baselines iCaRL | | D+R | | ZSCL | | LP-init-LDIFS (Ours) | |
|---|---|---|---|---|---|---|---|---|---|---|---|---|---|
| | | $\Delta_{LP}(\uparrow)$ | $\mathcal{A}_{LP}(\uparrow)$ | $\Delta_{LP}(\uparrow)$ | $\mathcal{A}_{LP}(\uparrow)$ | $\Delta_{LP}(\uparrow)$ | $\mathcal{A}_{LP}(\uparrow)$ | $\Delta_{LP}(\uparrow)$ | $\mathcal{A}_{LP}(\uparrow)$ | $\Delta_{LP}(\uparrow)$ | $\mathcal{A}_{LP}(\uparrow)$ | $\Delta_{LP}(\uparrow)$ | $\mathcal{A}_{LP}(\uparrow)$ |
| SVHN → C10 → R45 | SVHN | −3.81 | 90.48 | −3.21 | 91.9 | −3.67 | 91.62 | −2.78 | 93.3 | −3.23 | 92.7 | −0.41 | 96.68 |
| | CIFAR10 | −2.9 | 93.9 | −2.32 | 94.88 | −2.1 | 95.17 | −1.9 | 95.41 | −1.6 | 95.82 | −0.21 | 97.41 |
| | RESISC45 | 3.1 | 94.22 | 2.98 | 93.9 | 2.83 | 93.72 | 3.68 | 94.94 | 3.62 | 94.89 | 3.7 | 95.0 |
| | Others | −4.2 | 80.73 | −3.76 | 81.31 | −4.11 | 80.78 | −3.2 | 81.86 | −2.8 | 83.1 | 0.1 | 87.08 |
| SVHN → C100 → R45 | SVHN | −4.34 | 89.48 | −4.08 | 90.29 | −3.13 | 90.97 | −3.23 | 92.3 | −3.92 | 91.81 | −0.65 | 96.32 |
| | CIFAR100 | −3.25 | 83.24 | −3.01 | 83.95 | −3.13 | 84.06 | −2.6 | 84.82 | −2.13 | 85.07 | −0.3 | 86.54 |
| | RESISC45 | 3.21 | 93.8 | 3.62 | 94.91 | 3.54 | 94.87 | 3.71 | 95.08 | 3.65 | 94.96 | 3.83 | 95.11 |
| | Others | −4.11 | 81.73 | −3.8 | 82.04 | −4.02 | 81.62 | −3.43 | 82.17 | −3.11 | 82.86 | −0.23 | 89.12 |
| SVHN → Cars → R45 | SVHN | −3.64 | 91.43 | −2.92 | 92.74 | −3.13 | 91.75 | −2.84 | 92.86 | −2.72 | 92.98 | −0.17 | 96.9 |
| | Cars | −2.79 | 81.69 | −2.64 | 81.82 | −2.8 | 81.7 | −2.12 | 82.11 | −1.84 | 82.68 | 0.47 | 84.23 |
| | RESISC45 | 3.34 | 93.92 | 3.55 | 94.96 | 3.58 | 94.97 | 3.72 | 95.19 | 3.63 | 95.04 | 3.73 | 95.27 |
| | Others | −4.07 | 81.63 | −3.6 | 82.24 | −3.89 | 81.88 | −3.12 | 82.73 | −2.8 | 83.1 | 0.23 | 89.39 |

Table 4: $\Delta_{LP}$ and $\mathcal{A}_{LP}\%$ comparing LDIFS with 5 classic continual learning methods on the 3-task setup.

**Fine-tuning in foundation models**: There are many flavours of fine-tuning which improve on CLIP's zero-shot performance. A popular approach is prompt tuning where, methods like CoOp Zhou et al. (2022b), CoCoOp Zhou et al. (2022a), TPT Shu et al. (2022), Chain of Thought prompt tuning Ge et al. (2023), instead of hand-crafting prompts, learn the prompts specific to the fine-tuning task. Another category of methods uses adapters: CLIP-Adapter Gao et al. (2021), Tip-Adapter Zhang et al. (2021), Prompt Adapter Sun et al. (2023), SVL-Adapter Pantazis et al. (2022), which works on the principle of combining pre-trained features with a small non-linear adapter network where the adapter is fine-tuned on the downstream task at hand. Along with end-to-end fine-tuning, there are weight space interpolation methods like Wise-FT Wortsman et al. (2022b), PAINT Ilharco et al. (2022b), task arithmetic Ilharco et al. (2022a) and model soups Wortsman et al. (2022a) which look at interpolating between pre-trained and fine-tuned models in the weight space to achieve the best of both worlds in terms of downstream task performance and robustness to distribution shift.

In Table 8 of the appendix, we compare linear probing, CoOp, CLIP-Adapter, Tip-Adapter and the classic end-to-end fine-tuning method, ZS-init-CE on downstream task performance on 9 image classification tasks. Not only do we find end-to-end fine-tuning to consistently outperform adapters and prompt tuning, but when there is a big performance gap between the linear probe and end-to-end fine-tuning, adapters and prompt tuning do not bridge that gap well. This observation reaffirms the importance of end-to-end fine-tuning and thereby necessitates the investigation of better end-to-end fine-tuning methods for tasks where the pre-trained model's performance is sub-par. In Table 7 in the appendix, we also show how concept forgetting in end-to-end fine-tuning methods used in our work can be further improved by combining them with Wise-FT.

**Knowledge Distillation & Continual Learning**: The LDIFS regularizer can be studied from the lens of knowledge distillation, an approach which has been applied to different ends like calibration He et al. (2023), pre-training Lee et al. (2022), transfer learning Zhou et al. (2022c), robust fine-tuning Mao et al. (2022) and continual learning Li & Hoiem (2017); Jung et al. (2016); Rebuffi et al. (2017); Hou et al. (2018). The main difference between these works and ours is that unlike classic distillation methods which mainly use the last layer features Jung et al. (2016) or the logits Li & Hoiem (2017); Rebuffi et al. (2017) directly for distillation, we concatenate features from shallower layers in the network when computing the LDIFS regularizer. Furthermore, while works like Zagoruyko & Komodakis (2016); Passban et al. (2021); Passalis & Tefas (2018); Heo et al. (2019) develop indirect methods of distilling from intermediate feature representations including additional attention layers Zagoruyko & Komodakis (2016); Passban et al. (2021), probability

Figure 10: $\mathcal{A}_{LP}$ for models trained with LDIFS on EuroSAT using the full concatenated feature vector vs just the last layer (LL) feature vectors. Full feature vectors are crucial for LDIFS's performance.

distribution matching Passalis & Tefas (2018), and hidden neuron activation boundaries Heo et al. (2019), LDIFS uses L2 distance between intemediate feature representations directly as the regularizer. This makes it simpler to implement without any additional layers to be trained on intermediate feature representations.

Finally, the proposed modifications to distillation in LDIFS turn out to be crucial for performance. To investigate the importance of distilling from earlier features, in Figure 10, we compare performance of LDIFS when distilling from just last layer features (we call this ablation LL) during fine-tuning on EuroSAT. These plots provide evidence that using just the last layer features is not nearly as performant as using including the earlier features in the feature vector. We provide further insights on this in Appendix C.8. Finding which features contain pre-trained information for a downstream task is an interesting area of further research.

**LDIFS vs LPIPS Zhang et al. (2018)**: One can find similarities between our proposed LDIFS regulariser and the LPIPS metric Zhang et al. (2018) popularly used for measuring perceptual similarity between images. However, while LPIPS uses feature space distance on a *pre-trained, frozen* model to find perceptual similarity between pairs or sets of images, LDIFS instead feature space distance between a pair of pre-trained and fine-tuned model for the same image, to preserve the input-output behaviour of the pre-trained model.

## 7 Conclusions & Remarks

We explore how end-to-end fine-tuning approaches often applied on foundation models can, in turn, significantly damage the model's ability to recognize concepts outside the fine-tuning task at hand, a phenomenon which we call concept forgetting. Such an effect is undesirable particularly for foundation models which have been specifically trained on millions of samples to encode information on a vast number of real-world concepts. However, we find that among fine-tuning methods, L2SP, which keeps the model near the pre-trained model in its parameter space suffers less from concept forgetting. From insights gained by analyzing L2SP, we also find that feature space distance provides a better definition for the vicinity of the pre-trained model as the pre-trained concepts are indeed encoded in the feature space. Our proposed new regularizer, LDIFS, encourages the features of fine-tuned model to be in the vicinity of the pre-trained ones. Through extensive evaluation on 10 different downstream classification tasks, as well as a continual fine-tuning setup of 3 different sequences of 3 tasks each, we showed that LDIFS significantly alleviates concept forgetting without impeding downstream task performance.

**Limitations & Future work** Though we investigated concept forgetting and proposed a promising fix, we believe that various other experiments and analyses along these lines would be valuable for the community. First, LDIFS has an associated hyperparameter $\lambda_{LDIFS}$ which balances the objective of preserving pre-trained model behaviour with the objective of fine-tuning on the task at hand. Following convention, we cross-validate this hyperparameter on a held-out validation set (see Appendix A) from the task we fine-tune on. While this performs well, it is interesting to see if there can be better ways of setting this hyperparameter and in which cases the cross-validated hyperparameter may not yield the best model either at concept forgetting or downstream performance. Second, to compute LDIFS, we use a fixed set of features evenly spaced out in the network. In this regard, an analysis of which features are useful for encoding which concepts is still an open question worth investigating. Third, LDIFS tries to preserve the behaviour of the pre-trained model while fine-tuning. If there are biases and undesirable behaviours in the pre-trained model, these can percolate into the fine-tuned model. It is worth studying how such undesirable behaviours can be prevented from being distilled into the fine-tuned model while preserving the pre-trained knowledge. Fourth, understanding concept forgetting in other foundation model families such as Large Language Models, would be interesting. Works such as Jin et al. (2021); Scialom et al. (2022) have already explored this direction to a certain extent and it

would be interesting to identify commonalities and differences in fine-tuning approaches between different foundation model families. Finally, fundamentally defining what a "concept" is and the granularity at which we should study it for foundation models is important and requires research efforts to understand phenomena like "concept forgetting" beyond just performing task-based performance evaluation on a downstream dataset.

**Acknowledgements** This work is supported by the UKRI grant: Turing AI Fellowship EP/W002981/1 and EPSRC/MURI grant: EP/N019474/1. The authors would also like to thank the Royal Academy of Engineering and FiveAI.

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

## A    Training details

**Training datasets and hyper-parameters:** We fine-tune the CLIP ViT-B/32 model on each of the 10 image classification datasets: **a)** Stanford Cars, **b)** CIFAR-10, **c)** CIFAR-100, **d)** DTD, **e)** EuroSAT, **f)** GTSRB, **g)** MNIST, **h)** RESISC45, **i)** SVHN, **j)** ImageNet. For each dataset, we have a separate fine-tune run for each of the baselines discussed in §2. For each run, we train using the AdamW Loshchilov & Hutter (2017) optimizer, with an initial learning rate of $1e-5$, a weight decay of 0.1 and a cosine learning rate scheduler with a warmup length of 500. For all the runs, we use a batch-size of 128. Following the code of Ilharco et al. (2022a), we use the following number of epochs to fine-tune each dataset: **a)** Stanford Cars: 35 epochs, **b)** CIFAR-10/100: 10 epochs, **c)** DTD: 76 epochs, **d)** EuroSAT: 12 epochs, **e)** GTSRB: 11 epochs, **f)** MNIST: 10 epochs, **g)** RESISC45: 15 epochs, **h)** SVHN: 10 epochs and **j)** ImageNet: 10 epochs. We keep a minimum of 10 epochs for fine-tuning.

**Compute and number of experiments:** Each of our fine-tuning runs is done on a single NVIDIA A100 GPU. As there are a total of 10 classification datasets and a 8 fine-tuning baselines (including our proposed LDIFS), we perform a total of 80 fine-tune training runs from a pre-trained CLIP ViT-B/32. Furthermore, in order to produce the plots capturing $\mathcal{A}_{\mathrm{LP}}$, as well as $\ell_2$ distance in parameter and feature space, we store intermediate checkpoints over the course of fine-tuning. Particularly, for each run, we evaluate checkpoints after every 20% of training completion. Finally, in order to obtain the full set of results in Table 1 and Table 9, for the 72 models fine-tuned on datasets other than ImageNet, we evaluate each of them on 9 datasets (i.e., including the test set of the fine-tuned task), thereby having a total of 648 evaluations. For the 8 models fine-tuned on ImageNet, we evaluate them on a total of 8 datasets (leaving out CIFAR-10/100), thereby totalling 64 evaluations. Thus, we have a total of $648 + 64 = 712$ evaluation runs. Similar to training, each evaluation is also performed on a single NVIDIA A100 GPU.

**Choosing $\lambda_{\mathrm{LDIFS}}$:** One hyper-parameter which the LDIFS regularizer introduces is $\lambda_{\mathrm{LDIFS}}$. A higher value of $\lambda_{\mathrm{LDIFS}}$ encourages the model to preserve features of the original foundation model and vice versa. For each classification task, we performed a grid search over $\lambda_{\mathrm{LDIFS}} \in \{0.01, 0.05, 0.1, 0.5, 1, 10, 100\}$ and cross-validated this hyper-parameter on a held-out validation set, choosing the value which produces the best performance on the validation set. We found $\lambda_{\mathrm{LDIFS}} = 10$ to produce the best performance over datasets in general, so all the results we present in this paper are with $\lambda_{\mathrm{LDIFS}}$ set to 10.

## B    Computing $\Phi_\theta(\mathbf{x})$

In this section, we discuss how we compute the concatenated feature vector $\Phi_\theta(\mathbf{x})$ given input $\mathbf{x}$ and model parameters $\theta$, specifically for a ViT-B/32 model. This is used for computing LDIFS, our proposed regularizer for fine-tuning.

Let the feature output from layer $l$ in the network for input $\mathbf{x}$ be $\Phi_{\theta(l)}(\mathbf{x}) \in \mathbb{R}^l$. Then the normalized feature vector for layer $l$ can be represented as $\frac{\Phi_{\theta(l)}(\mathbf{x})}{||\Phi_{\theta(l)}(\mathbf{x})||}$. In order to form the concatenated feature vector, one can take technically take features from every intermediate layer of the network. However, storing all the features is memory intensive. Thus, we follow a similar approach to the LPIPS Zhang et al. (2018) implementation for VGG and AlexNet, and choose 5 intermediate points in the ViT-B/32 architecture to collect features from. For a single input image $\mathbf{x} \in \mathbb{R}^{3 \times 224 \times 224}$, each of the intermediate feature representations has a dimension of $\mathbb{R}^{50 \times 768}$ with 50 tokens and 768 dimensional representation for each token. When normalizing the feature vector, we flatten this vector out to a single 38400 dimensional vector. Thus the full concatenated feature vector $\Phi_\theta(\mathbf{x})$ has dimensionality $5 \times 38400 = 192000$. However, note that there can be other ablations to this design and we leave that for future exploration.

# C   Additional results

In this section, we present additional empirical results to supplement our observations and conclusions in the main paper.

## C.1   Concept forgetting with $\mathcal{A}_{ZS}$ and $\Delta_{ZS}$

In Section 3.1, we proposed a simple method to quantify concept forgetting using the difference in LP accuracy $\Delta_{LP}$ between the pre-trained and fine-tuned models on a task. We also discussed how using ZS accuracy $\mathcal{A}_{ZS}$ in the same way may not be suitable, as ZS linear classifiers may not be indicative of a model recognizing concepts post fine-tuning. Nonetheless, for the sake of completeness of our analysis, in the additional results we present below, we also include $\mathcal{A}_{ZS}$ and compute $\Delta_{ZS}$ as:

$$\Delta_{ZS}(\mathcal{D}, f_{\theta_v}, f_{\theta_{v(0)}}) = \mathcal{A}_{ZS}(f_{\theta_v}, \mathcal{D}) - \mathcal{A}_{ZS}(f_{\theta_{v(0)}}, \mathcal{D}). \tag{6}$$

Similar to $\Delta_{LP}$, a negative $\Delta_{ZS}$ would indicate concept forgetting, a zero would indicate knowledge accumulation and a positive value would indicate knowledge gain or positive forward transfer on the task under inspection.

## C.2   Crippling effect of fine-tuning

In §3, we observed how most existing fine-tuning methods, while gaining state-of-the-art performance on fine-tuned tasks, can lead to concept forgetting in models. In this section, we provide additional empirical results to further strengthen those observations.

In Figure 11 and Figure 17, we present the $\mathcal{A}_{ZS}$ and $\mathcal{A}_{LP}$ accuracy respectively, for models fine-tuned on 9 classification tasks, on their respective test sets. This shows expected behaviour as models broadly achieve very high test set accuracy, which increases over the course of fine-tuning, on their respective fine-tuned tasks. Interestingly, LP-init baselines seem to have relatively lower performance on $\mathcal{A}_{ZS}$ accuracy compared to their ZS-init counter-parts. However, this reduced performance is only limited to $\mathcal{A}_{ZS}$ as this does not translate to a reduced performance in $\mathcal{A}_{LP}$. Moreover, L2SP baselines (both ZS and LP-init) obtain relatively lower fine-tuned accuracy both in case of $\mathcal{A}_{ZS}$ and $\mathcal{A}_{LP}$.

In Figure 12 and Figure 18, we present the $\mathcal{A}_{ZS}$ and $\mathcal{A}_{LP}$ for models fine-tuned on EuroSAT (first row), GTSRB (second row) and SVHN (third row) as captured on 8 classification datasets different from their respective fine-tuned set. Broadly, the ZS and LP performance for all models drops on other datasets as they are fine-tuned, thereby capturing concept forgetting in the fine-tuned models. Among the baselines, we consistently observe LP-init-L2SP to perform better than others in avoiding concept forgetting. This is evident through the distinctly higher $\mathcal{A}_{ZS}$ and $\mathcal{A}_{LP}$ accuracies over fine-tuning for the LP-init-L2SP baselines.

## C.3   Analysing L2SP

In this section, we provide additional results to supplement our observations related to investigating the L2SP baseline in §4 of the main paper. In Figure 21, we present the $\ell_2$ distance in parameter space $||\theta - \theta_0||_2^2$ between the pre-trained and current models captured over fine-tuning. For all the datasets, we observe that L2SP baselines while initially diverging slightly in the parameter space, converge back to a model having low $\ell_2$ parameter space distance from the original foundation model. Other baselines on the other hand, completely diverge away from the original model in the parameter space.

To further investigate the change in input-output behaviour of the model over fine-tuning, we measure the distance in feature space (see Equation (3)) over fine-tuning. In Figure 22, we present the $\ell_2$ distance in feature space captured for models fine-tuned on EuroSAT. Again, consistent with our previous observation, we find that unlike other baselines, L2SP first diverges away from the original foundation model and then converges back to the original input-output behaviour, as is indicative through a decreasing L2 feature space distance in the later stages of fine-tuning. This observation is consistent on the EuroSAT train set, the EuroSAT test set as well as on other datasets, thereby providing our motivation for the LDIFS regularizer.

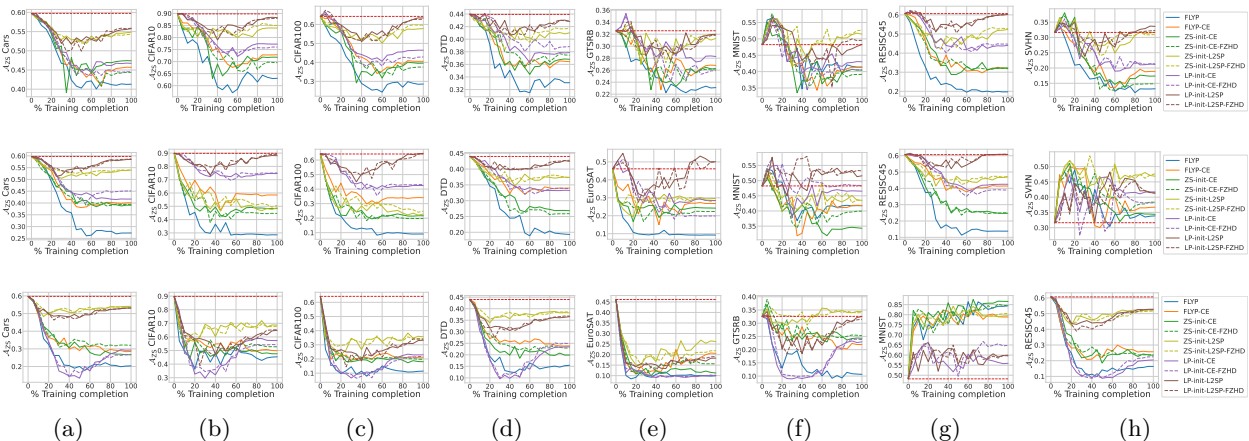

Figure 11: **Test set ZS Accuracy $\mathcal{A}_{ZS}$ for different fine-tuning methods** on 9 image classification tasks: (a) Stanford Cars, (b) CIFAR-10, (c) CIFAR-100, (d) DTD, (e) EuroSAT, (f) GTSRB, (g) MNIST (h) RESISC45 and (i) SVHN. $\mathcal{A}_{ZS}$ generally rises over the course of fine-tuning. However, for the ZS-init-L2SP and LP-init-L2SP baselines, the gain in $\mathcal{A}_{ZS}$ is relatively lower. Furthermore, for LP-init baselines, the performance is consistently lower compared to other baselines

Figure 12: **ZS Accuracy $\mathcal{A}_{ZS}$ for models fine-tuned on EuroSAT (first row), GTSRB (second row), and SVHN (third row) and evaluated on 8 different datasets different from their fine-tuning dataset.** Most fine-tuning methods show a drop in $\mathcal{A}_{ZS}$ performance over the course of fine-tuning indicating a reduction in the model's transferability.

### C.4 Performance of LDIFS

In this section, we provide additional results to supplement observations related to the performance of the proposed LDIFS regularizer.

Firstly, in Figure 15 and Figure 19, we present the test set accuracy of all fine-tuning methods including LDIFS on all 9 classification tasks. We see that LDIFS performs competitively with all other baselines and improves on L2SP consistently. In Figure 16 and Figure 20, we present the $\mathcal{A}_{ZS}$ and $\mathcal{A}_{LP}$ accuracy respectively, for models fine-tuned on EuroSAT (first row), GTSRB (second row) and SVHN (third row) over the course of fine-tuning. Furthermore, we also provide the $\Delta_{ZS}$, $\Delta_{LP}$ and the $\mathcal{A}_{LP}$ values for fully fine-tuned models on 9 classification tasks and 8 baselines in Table 9.

As is clear from our results, LP-init-LDIFS consistently minimizes concept forgetting without compromising on performance on the fine-tuned task. This is evident from its high $\mathcal{A}_{ZS}$, $\mathcal{A}_{LP}$ (see Figure 16 and Figure 20) and high $\Delta_{ZS}$ and $\Delta_{LP}$ (see Table 9) when evaluated on other tasks. Note from Table 9 that even when LP-init-LDIFS is not achieving the best performance on a certain dataset pair, it is often a close second with very little difference compared to the best performing baseline. In addition to this advantage, its $\mathcal{A}_{LP}$ accuracy on the fine-tuned task itself is very competitive with the top scores obtained by other fine-tuning baselines and provides a consistent improvement on L2SP.

### C.5 Comparison with Adapters and Prompt Tuning

In Table 8, we present results of fine-tuning a CLIP ViT-B/32 model using linear probing, CoOp, CLIP-Adapter, Tip-Adapter and full end-to-end fine-tuning using ZS-init-CE loss on 9 image classification tasks.

We consistently observe end-to-end fine-tuning to produce the best test set accuracy across all 9 tasks. Furthermore, we also note that in case of tasks where the gap between linear probe performance and end-to-end fine-tuning performance is significant (e.g. SVHN), adapter and prompt tuning approaches don't cover this performance gap. Hence, if the pre-trained encoder is not performant on a downstream task, the only way to achieve state-of-the-art results on the task requires end-to-end fine-tuning the model on the task itself. This observation thus necessitates study into better end-to-end fine-tuning methods for foundation models.

### C.6    Comparison with Wise-FT

In this section, we perform an ablation to see the effect of Wise-FT Wortsman et al. (2022b) on preventing concept forgetting. Wise-FT is a weight interpolation method which forms a linear combination between the pre-trained $\theta_0$ and fine-tuned $\theta_f$ model parameters:

$$\theta_{\text{wse}} = \alpha\theta_0 + (1 - \alpha)\theta_f \tag{7}$$

Since this can work with any fine-tuned model, we can apply it on all the end-to-end fine-tuning baselines in this work. Specifically, we tune the hyperparameter $\alpha$ on a held-out validation set of the downstream task at hand to maximum validation accuracy. We evaluate concept forgetting on 3 downstream tasks: CIFAR-10, EuroSAT and SVHN using the $\Delta_{\text{LP}}$ metric and report the results in Table 7. For each dataset, we evaluate the mean $\Delta_{\text{LP}}$ on 5 other downstream tasks: Cars, DTD, GTSRB, RESISC45 and MNIST. Our observations show a consistent reduction in concept forgetting across all fine-tuning baselines. However, the order of performance between the fine-tuning baselines does not change and LDIFS still maintains superior performance over other baselines both pre and post application of Wise-FT. Thus, the results show that Wise-FT can be combined with any E2E fine-tuning method to further minimize concept forgetting.

### C.7    Additional results on continual fine-tuning

**Ablation with joint training.** Joint training Li & Hoiem (2017) is a continual learning baseline often used to upper bound the performance of continual learners. The idea is to use samples from different tasks in a manner that minimizes forgetting. In this experiment, we take the simple setting of joint training, where at each step of the sequence when a new dataset comes in, we retrain the model on the combination of all the training data from earlier datasets in the sequence along with the new incoming data. The training is done at each point using the ZS-init-CE loss function, which is the vanilla go-to fine-tuning method.

In Table 5, we compare results of the joint training baseline with LP-init-LDIFS. Note that $\Delta_{\text{LP}}$ is computed using Equation (5) where we compute forgetting on a task in the sequence as the difference between the model fine-tuned on the entire sequence and the model in the sequence which performed the best (had the most knowledge) on the task. We can make the following observations from the results:

1. **Earlier tasks in the sequence:** On tasks seen earlier in the sequence, (like SVHN), LP-init-LDIFS is competitive but slightly underperforms compared to joint training which sees the full training set comprising all previous tasks at every point in the sequence.
2. **Current/Last task in the sequence:** On the final task in the sequence, i.e., RESISC45, LDIFS performs competitively with the joint training baseline.
3. **Tasks not in the sequence:** On tasks not seen in the fine-tuning sequence, LDIFS significantly outperforms joint training exhibiting its ability to minimize concept forgetting.

**Comparison with continual learning baselines that use pre-trained encoders.** While the motivation of the work lies in fine-tuning foundation models on downstream tasks, and in their eventual continual fine-tuning, the proposed LDIFS method is general enough to be applied in standard continual learning setups as well. Furthermore, in the continual learning literature, there are works like L2P Wang et al. (2022c), DualPrompt Wang et al. (2022b), CODA-Prompt Smith et al. (2023), Continual-CLIP Thengane et al. (2022) which utilize pre-training in order to improve performance on well-known continual learning

| Dataset Fine-tune | Eval | Continual Learning baselines | | | |
|---|---|---|---|---|---|
| | | Joint Training | | LP-init-LDIFS (Ours) | |
| | | $\Delta_{LP}(\uparrow)$ | $\mathcal{A}_{LP}(\uparrow)$ | $\Delta_{LP}(\uparrow)$ | $\mathcal{A}_{LP}(\uparrow)$ |
| SVHN → C10 → R45 | SVHN | −0.02 | 97.04 | −0.41 | 96.68 |
| | CIFAR10 | −0.1 | 97.76 | −0.21 | 97.41 |
| | RESISC45 | 3.88 | 95.19 | 3.84 | 95.0 |
| | Others | −4.74 | 80.13 | 0.1 | 87.08 |
| SVHN → C100 → R45 | SVHN | −0.23 | 96.77 | −0.65 | 96.32 |
| | CIFAR100 | 0.05 | 87.02 | −0.3 | 86.54 |
| | RESISC45 | 3.86 | 95.07 | 3.83 | 95.11 |
| | Others | −4.67 | 80.62 | −0.23 | 89.12 |
| SVHN → Cars → R45 | SVHN | −0.11 | 97.13 | −0.17 | 96.9 |
| | Cars | 0.32 | 84.12 | 0.47 | 84.23 |
| | RESISC45 | 3.83 | 95.24 | 3.83 | 95.27 |
| | Others | −4.95 | 83.42 | 0.23 | 89.39 |

Table 5: $\Delta_{LP}$ and $\mathcal{A}_{LP}\%$ comparing LDIFS with joint training as a CL baseline on the 3-task setup.

| | | | Average accuracy | | |
|---|---|---|---|---|---|
| L2P | DualPrompt | CODA-Prompt | Continual-CLIP | SLCA | LP-init-LDIFS (ours) |
| $74.6 \pm 1.21$ | $77.24 \pm 1.27$ | $78.13 \pm 1.18$ | $76.23 \pm 1.18$ | $81.22 \pm 1.23$ | $\mathbf{83.62 \pm 1.16}$ |

Table 6: Performance of continual learning baselines on Split ImageNet-R.

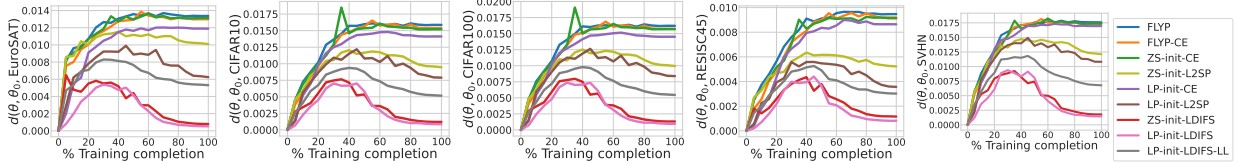

Figure 13: $l_2$ **distance in between features obtained from a pre-trained model and a model fine-tuned on EuroSAT.** The last-layer ablation for LDIFS does not minimize feature space distance compared to using earlier features as well.

benchmarks. While we don't target our method specifically towards such benchmarks, in this section, we provide a comparison on Split ImageNet-R Wang et al. (2022b), a well-known class-incremental learning benchmark. Specifically, we compare with the other methods mentioned above which use pre-training as part of their method to improve continual learning. We present results in Table 6. It is clear from the results that LP-init-LDIFS outperforms competitive baselines indicating its usability in classic class-incremental learning benchmarks as well.

### C.8 Last Layer Ablation

In this section, we present some results to provide additional insights on why LDIFS using just the last layer does not perform as well as using the full feature vector. To see this, we plot the feature space distance $d(\theta, \theta_0, \mathcal{D})$ for all baselines including LDIFS as well as LP-init-LDIFS with just the last layer for models fine-tuned on EuroSAT. The plots are shown in Figure 13. These plots show that the full feature distance is not minimized if we just use last layer feature difference in the LDIFS regularizer, thereby indicating that learned concepts can indeed be encoded in shallower layers of the network which makes distilling from these layers crucial.

### C.9 Visualizing Table 1

In this section, we present the results of Table 1 using an alternate visualization. For 9 downstream datasets, we plot the LP accuracy on the dataset itself on the x-axis and the average LP accuracy on all other datasets on the y-axis. The best baselines should lie on the top right corner of this plot as that indicates that as the model is fine-tuned on the dataset, it still retains performance on other datasets. Results are in Figure 14. Clearly, we consistently see the LDIFS baselines to lie at the top right, again indicative of their performance in not just minimizing concept forgetting but also obtaining excellent downstream task performance.

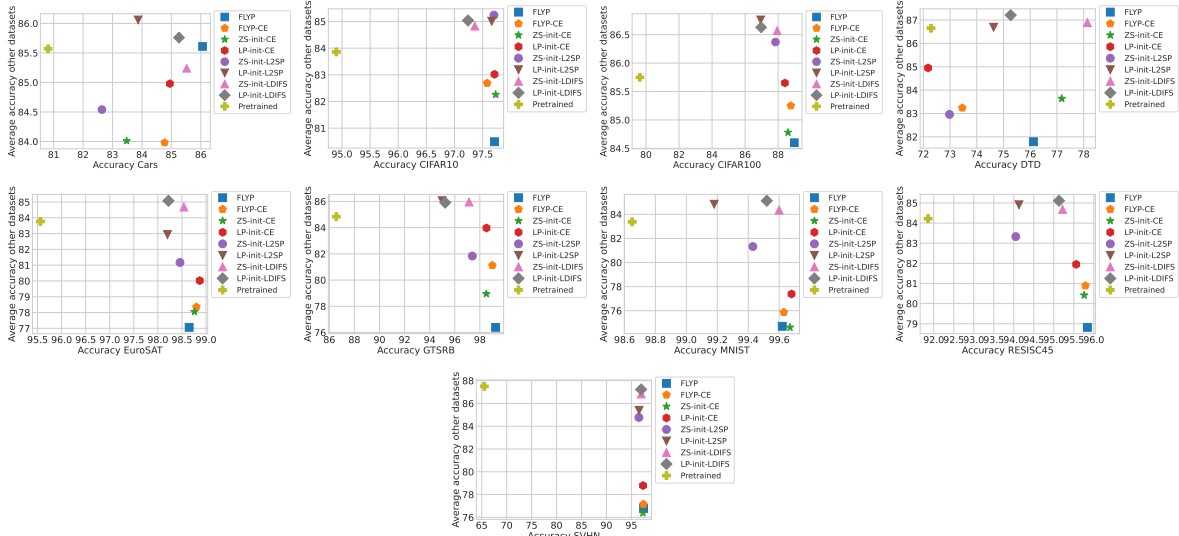

Figure 14: **Visualizing test LP accuracy of fine-tuning baselines on different datasets.** For each dataset, we plot its test set LP accuracy on the x-axis and the average test set LP accuracy of the remaining datasets on the y-axis.

| Dataset | Fine-tuning baselines | | | | | | | | | | | | | |
|---|---|---|---|---|---|---|---|---|---|---|---|---|---|---|
| | FLYP | | FLYP-CE | | ZS-init-CE | | LP-init-CE (LP-FT) | | ZS-init-L2SP | | LP-init-L2SP | | ZS-init-LDIFS (Ours) | | LP-init-LDIFS (Ours) |
| | $\Delta_{LP}$ | $\Delta_{LP}(+wse)$ | $\Delta_{LP}$ | $\Delta_{LP}(+wse)$ | $\Delta_{LP}$ | $\Delta_{LP}(+wse)$ | $\Delta_{LP}$ | $\Delta_{LP}(+wse)$ | $\Delta_{LP}$ | $\Delta_{LP}(+wse)$ | $\Delta_{LP}$ | $\Delta_{LP}(+wse)$ | $\Delta_{LP}$ | $\Delta_{LP}(+wse)$ | $\Delta_{LP}$ | $\Delta_{LP}(+wse)$ |
| CIFAR10 | −5.14 | −0.11 | −1.33 | −0.16 | −1.67 | −0.1 | −1.49 | −0.01 | 1.58 | 1.64 | 1.49 | 1.96 | 1.53 | 1.99 | **1.72** | **2.01** |
| EuroSAT | −6.5 | −1.32 | −5.53 | −1.04 | −5.62 | −1.23 | −4.08 | −0.76 | −1.77 | −0.12 | −0.87 | 0.08 | 0.83 | 1.2 | **1.24** | **1.96** |
| SVHN | −10.82 | −6.24 | −10.18 | −6.37 | −10.98 | −7.2 | −9.2 | −4.11 | −2.65 | −0.66 | −2.13 | −0.23 | −0.57 | 0.12 | **−0.18** | **0.33** |

Table 7: $\Delta_{LP}$ without and with Wise-FT Wortsman et al. (2022b) for all fine-tuning baselines on CIFAR-10, EuroSAT and SVHN.

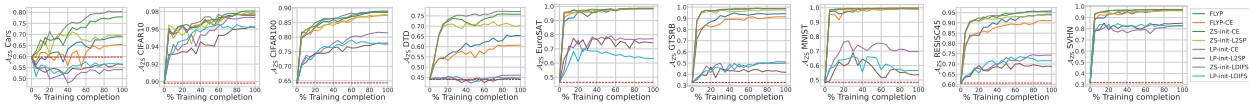

Figure 15: **Comparing test set ZS Accuracy $\mathcal{A}_{ZS}$ for different fine-tuning methods with LDIFS** on 9 image classification tasks. While LP-init baselines generally underperform on $\mathcal{A}_{ZS}$, we find ZS-init-LDIFS to be competitive with the best baselines.

| Dataset | LP | CoOp | CLIP-Adapter | Tip-Adapter | E2E Fine-tuning |
|---|---|---|---|---|---|
| Cars | 80.80 | 80.88 | 81.24 | 81.32 | **83.48** |
| CIFAR10 | 94.92 | 95.02 | 94.87 | 95.13 | **97.73** |
| CIFAR100 | 79.62 | 80.12 | 80.06 | 80.82 | **88.6** |
| DTD | 72.08 | 72.01 | 71.87 | 72.62 | **77.18** |
| EuroSAT | 95.56 | 95.14 | 95.38 | 96.23 | **98.76** |
| GTSRB | 86.70 | 86.81 | 87.35 | 88.04 | **98.52** |
| MNIST | 98.65 | 98.84 | 99.03 | 99.11 | **99.67** |
| RESISC45 | 91.86 | 91.79 | 91.82 | 91.80 | **95.76** |
| SVHN | 65.47 | 67.28 | 69.76 | 69.23 | **97.3** |

Table 8: Test set accuracy obtained from linear probing (LP), CoOp, CLIP-Adapter, Tip-Adapter and end-to-end (E2E) fine-tuning using ZS-init-CE loss.

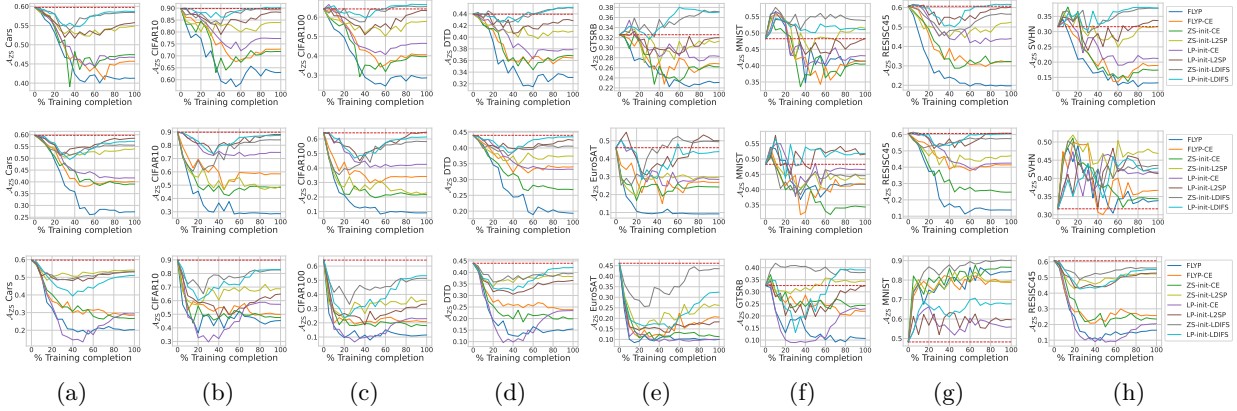

Figure 16: **Comparing ZS Accuracy $\mathcal{A}_{ZS}$ of different fine-tuning methods with LDIFS for models fine-tuned on EuroSAT (first row), GTSRB (second row) and SVHN (third row) and evaluated on 8 datasets different from their fine-tuning dataset.** LP-init-LDIFS almost consistently beats all other baselines including LP-init-L2SP in preserving the model's transferability.

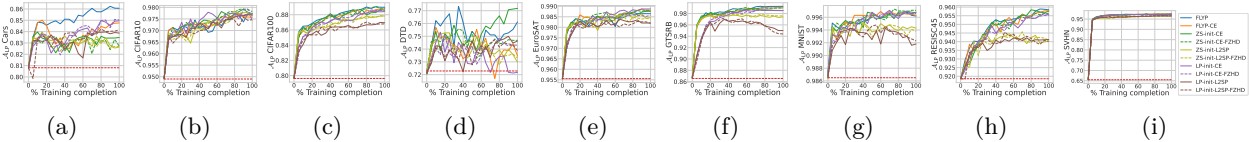

Figure 17: **Test set LP Accuracy $\mathcal{A}_{LP}$ for different fine-tuning methods** on 9 image classification tasks: (a) Stanford Cars, (b) CIFAR-10, (c) CIFAR-100, (d) DTD, (e) EuroSAT, (f) GTSRB, (g) MNIST (h) RESISC45 and (i) SVHN. $\mathcal{A}_{LP}$ generally rises over the course of fine-tuning. However, for the ZS-init-L2SP and LP-init-L2SP baselines, the gain in $\mathcal{A}_{LP}$ is relatively lower. Furthermore, for LP-init baselines, the performance is consistently lower compared to other baselines

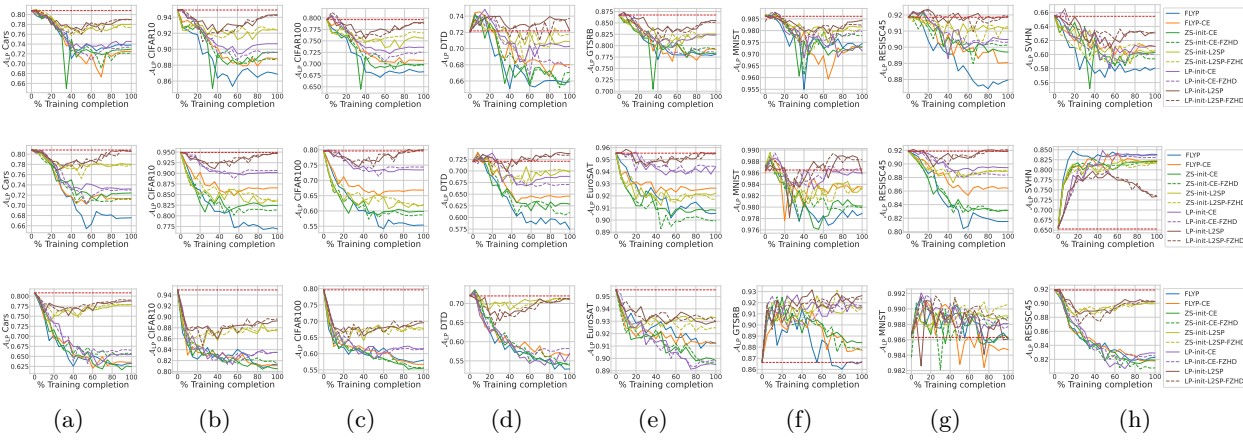

Figure 18: **LP Accuracy $\mathcal{A}_{LP}$ for models fine-tuned on EuroSAT (first row), GTSRB (second row), and SVHN (third row) and evaluated on 8 different datasets different from their fine-tuning dataset.** Most fine-tuning methods show a drop in $\mathcal{A}_{LP}$ performance over the course of fine-tuning indicating a reduction in the model's transferability.

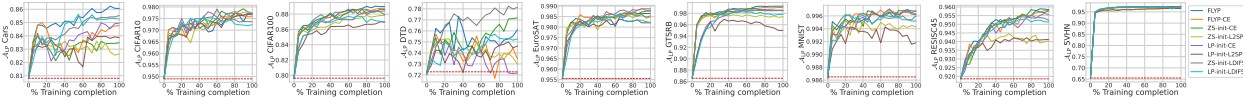

Figure 19: **Comparing test set LP Accuracy** $\mathcal{A}_{\mathrm{LP}}$ **for different fine-tuning methods with LDIFS on 9 image classification tasks.**

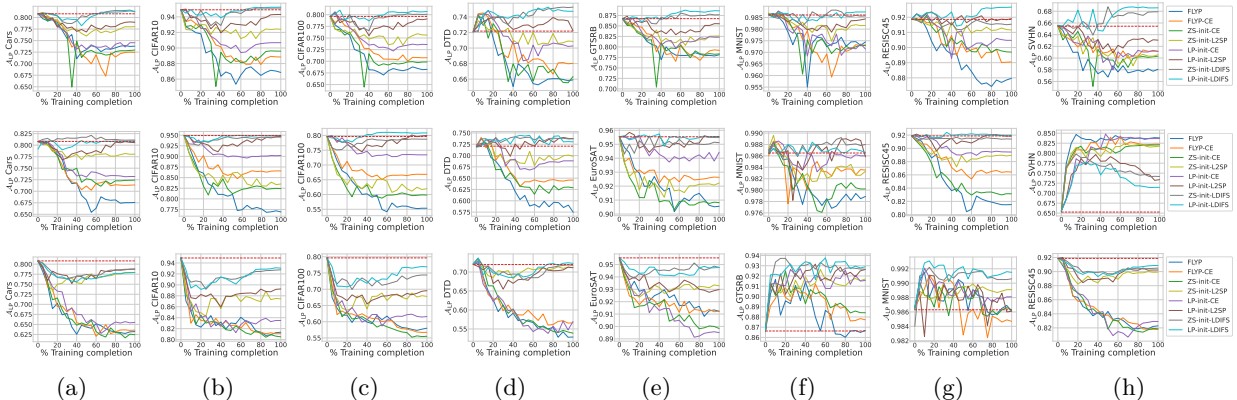

Figure 20: **Comparing LP Accuracy** $\mathcal{A}_{\mathrm{LP}}$ **of different fine-tuning methods with LDIFS for models fine-tuned on EuroSAT (first row), GTSRB (second row) and SVHN (third row) and evaluated on 8 datasets different from their fine-tuning dataset.** LP-init-LDIFS almost consistently beats all other baselines including LP-init-L2SP in preserving the model's transferability.

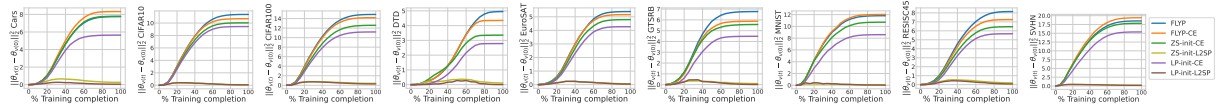

Figure 21: $\ell_2$ **distance in weight space** $||\theta - \theta_0||_2^2$ between pre-trained image encoder $f_{\theta_0}$ and fine-tuned image encoder $f_\theta$ over the course of fine-tuning. Apart from ZS-init-L2SP and LP-init-L2SP, all fine-tuning baselines diverge in weight space over the course of fine-tuning.

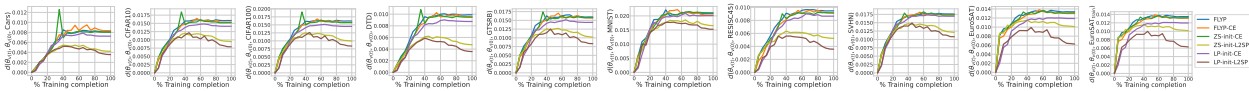

Figure 22: $\ell_2$ **distance in feature space** $d(\theta, \theta_0, \mathcal{D})$, between fine-tuned and pre-trained image encoders, $f_\theta$ and $f_{\theta_0}$ computed over different fine-tuning methods for models fine-tuned on EuroSAT.

| Fine-tune | Eval | FLYP $\Delta_{ZS}$ | $\Delta_{LP}$ | $\mathcal{A}_{LP}$ | FLYP-CE $\Delta_{ZS}$ | $\Delta_{LP}$ | $\mathcal{A}_{LP}$ | ZS-init-CE $\Delta_{ZS}$ | $\Delta_{LP}$ | $\mathcal{A}_{LP}$ | LP-init-CE (LP-FT) $\Delta_{ZS}$ | $\Delta_{LP}$ | $\mathcal{A}_{LP}$ | ZS-init-L2SP $\Delta_{ZS}$ | $\Delta_{LP}$ | $\mathcal{A}_{LP}$ | LP-init-L2SP $\Delta_{ZS}$ | $\Delta_{LP}$ | $\mathcal{A}_{LP}$ | ZS-init-LDIFS (Ours) $\Delta_{ZS}$ | $\Delta_{LP}$ | $\mathcal{A}_{LP}$ | LP-init-LDIFS (Ours) $\Delta_{ZS}$ | $\Delta_{LP}$ | $\mathcal{A}_{LP}$ |
|---|---|---|---|---|---|---|---|---|---|---|---|---|---|---|---|---|---|---|---|---|---|---|---|---|---|
| Cars | CIFAR10 | -3.55 | -0.68 | 94.24 | -9.0 | -1.74 | 93.18 | -9.81 | -1.52 | 93.4 | -10.42 | -0.93 | 94.0 | -2.78 | -0.89 | 94.02 | -0.26 | 0.06 | 94.97 | -2.98 | -0.47 | 94.45 | -2.85 | -0.22 | 94.69 |
| | CIFAR100 | -6.91 | -0.76 | 78.89 | -7.41 | -2.74 | 76.91 | -8.62 | -3.58 | 76.04 | -6.84 | -2.12 | 77.51 | -3.4 | -2.28 | 77.34 | 0.22 | 0.17 | 79.81 | -1.6 | -0.37 | 79.26 | -1.76 | 0.07 | 79.71 |
| | DTD | -2.61 | 1.28 | 73.46 | -4.2 | -1.44 | 70.9 | -3.99 | 0.53 | 72.5 | -3.99 | 2.13 | 74.26 | -3.09 | -0.64 | 71.44 | -1.17 | 2.39 | 74.63 | -1.54 | -0.32 | 71.76 | -2.34 | 0.43 | 72.55 |
| | EuroSAT | -16.94 | -0.2 | 95.33 | -5.89 | -0.54 | 95.0 | -6.69 | -0.8 | 94.74 | 2.74 | -0.02 | 95.52 | -4.8 | -0.56 | 94.98 | -2.46 | 0.02 | 95.56 | 1.0 | -0.13 | 95.41 | 4.41 | 0.02 | 95.56 |
| | GTSRB | -3.62 | 0.55 | 87.09 | -4.28 | -1.39 | 85.24 | -1.1 | -0.66 | 85.96 | -3.58 | -1.01 | 85.62 | 2.19 | -0.4 | 86.21 | 1.85 | 0.64 | 87.21 | 2.36 | -0.02 | 86.56 | 0.74 | 0.44 | 87.18 |
| | MNIST | 4.28 | -0.19 | 98.56 | 2.26 | -0.52 | 98.13 | 4.29 | -0.43 | 98.24 | 2.21 | -0.14 | 98.48 | 1.46 | -0.34 | 98.33 | 2.45 | -0.07 | 98.59 | 2.33 | 0.01 | 98.71 | 2.15 | 0.06 | 98.62 |
| | RESISC45 | -6.19 | -1.21 | 90.67 | -7.16 | -1.57 | 90.3 | -10.7 | -2.13 | 89.75 | -4.81 | -0.87 | 91.0 | -4.48 | -1.4 | 90.48 | -0.21 | -0.25 | 91.62 | -4.13 | -0.48 | 91.4 | -2.7 | -0.13 | 91.75 |
| | SVHN | 0.33 | 1.14 | 66.63 | -5.4 | -3.38 | 62.15 | -1.63 | -3.91 | 61.47 | -0.7 | -2.09 | 63.49 | 2.93 | -2.01 | 63.51 | 1.54 | 0.78 | 66.08 | -0.05 | -1.13 | 64.35 | 1.03 | 0.53 | 66.01 |
| | Cars | 9.2 | 5.25 | 86.06 | 5.73 | 3.95 | 84.77 | 18.24 | 2.67 | 83.48 | 9.44 | 4.14 | 84.95 | 9.44 | 1.83 | 82.64 | -3.31 | 3.06 | 83.87 | 20.52 | 4.71 | 85.52 | -3.15 | 4.45 | 85.26 |
| CIFAR10 | Cars | -58.0 | -14.58 | 66.24 | -45.32 | -7.04 | 73.75 | -46.08 | -5.68 | 75.12 | -36.86 | -8.27 | 72.53 | -15.81 | 0.52 | 81.32 | -4.19 | -0.55 | 80.26 | -13.78 | 0.53 | 81.33 | -3.95 | -1.31 | 79.51 |
| | CIFAR100 | -19.22 | 0.97 | 80.62 | -0.33 | 1.86 | 81.52 | -2.39 | 1.86 | 81.48 | 0.23 | 2.85 | 82.46 | 8.79 | 4.81 | 84.45 | 8.97 | 5.22 | 84.85 | 3.08 | 0.28 | 79.92 | 3.31 | 2.12 | 81.74 |
| | DTD | -14.31 | -7.93 | 64.15 | -9.52 | -4.52 | 67.71 | -9.73 | -6.06 | 66.01 | -9.52 | -4.36 | 67.77 | -1.86 | 1.38 | 73.46 | -2.77 | -0.74 | 71.33 | -0.8 | 0.27 | 72.34 | -0.69 | 0.05 | 72.18 |
| | EuroSAT | -16.93 | -0.07 | 95.46 | -14.76 | -0.3 | 95.24 | -18.06 | -0.19 | 95.35 | -17.04 | -0.33 | 95.2 | 4.2 | -0.07 | 95.46 | -1.09 | 0.44 | 95.98 | 0.17 | 0.07 | 95.61 | 2.83 | 0.46 | 96.0 |
| | GTSRB | -2.72 | -1.24 | 85.3 | -0.8 | 0.35 | 87.05 | -2.16 | -0.48 | 86.25 | -0.33 | 1.03 | 87.63 | 1.63 | 1.15 | 87.82 | -0.87 | 1.58 | 88.23 | 5.81 | 2.94 | 89.45 | 5.38 | 3.21 | 89.99 |
| | MNIST | -9.31 | -0.45 | 98.23 | -10.42 | -0.07 | 98.59 | -12.21 | -0.1 | 98.54 | -4.02 | 0.22 | 98.73 | 2.74 | -0.1 | 98.55 | 2.25 | -0.13 | 98.52 | -0.4 | 0.04 | 98.62 | 0.21 | -0.17 | 98.49 |
| | RESISC45 | -26.32 | -4.83 | 87.05 | -14.44 | -2.54 | 89.33 | -15.49 | -2.51 | 89.37 | -16.32 | -1.97 | 89.9 | -0.29 | -0.25 | 91.62 | -1.49 | -0.1 | 91.78 | 0.25 | 0.17 | 92.05 | 1.03 | 0.13 | 92.0 |
| | SVHN | -3.49 | 1.36 | 66.84 | -7.19 | 2.9 | 68.33 | -5.45 | 0.4 | 65.98 | 0.62 | 4.34 | 69.95 | 7.15 | 3.71 | 69.22 | 7.08 | 3.59 | 69.13 | 6.17 | 3.97 | 69.34 | 8.13 | 4.92 | 70.4 |
| | CIFAR10 | 7.87 | 2.79 | 97.71 | 7.8 | 2.65 | 97.58 | 8.25 | 2.83 | 97.73 | 7.5 | 2.79 | 97.71 | 8.05 | 2.8 | 97.7 | 6.38 | 2.74 | 97.66 | 7.72 | 2.46 | 97.36 | 6.39 | 2.33 | 97.24 |
| CIFAR100 | Cars | -27.76 | -3.15 | 77.66 | -12.57 | -3.36 | 77.44 | -21.22 | -3.58 | 77.23 | -12.42 | -4.38 | 76.42 | -6.74 | -0.12 | 80.69 | -1.32 | 0.15 | 80.96 | -12.29 | -0.72 | 80.08 | -6.32 | -1.07 | 79.73 |
| | CIFAR10 | -8.17 | 1.05 | 95.96 | 0.22 | 1.33 | 96.25 | -0.64 | 1.5 | 96.41 | 1.06 | 1.52 | 96.42 | 2.65 | 1.96 | 96.87 | 3.24 | 1.62 | 96.53 | 0.33 | 1.34 | 96.25 | 0.43 | 0.68 | 95.79 |
| | DTD | -10.37 | -6.44 | 65.64 | -8.83 | -5.53 | 66.54 | -9.36 | -5.11 | 66.97 | -6.54 | -3.3 | 68.83 | -4.52 | -0.74 | 71.33 | -2.39 | 0.16 | 72.23 | -4.41 | -1.44 | 70.64 | -1.97 | -1.97 | 70.11 |
| | EuroSAT | -23.19 | -0.06 | 95.48 | | -0.26 | 95.28 | -16.59 | -0.48 | 95.06 | -7.11 | -0.17 | 95.37 | -0.46 | 0.19 | 95.72 | | 0.46 | 96.0 | -3.33 | 0.26 | 95.8 | -2.19 | 0.56 | 96.09 |
| | GTSRB | -6.46 | 0.17 | 86.79 | -0.43 | 2.52 | 89.11 | -1.64 | 0.99 | 87.62 | 2.98 | 2.69 | 89.3 | 1.27 | 1.8 | 88.42 | 3.2 | 2.65 | 89.18 | 4.88 | 2.98 | 89.65 | 4.04 | | 90.7 |
| | MNIST | -8.99 | -0.35 | 98.36 | -7.26 | -0.15 | 98.51 | -1.37 | -0.3 | 98.36 | -6.36 | 0.11 | 98.79 | 3.09 | 0.08 | 98.52 | 2.85 | 0.04 | 98.75 | 8.1 | -0.07 | 98.68 | 0.25 | 0.2 | 98.95 |
| | RESISC45 | -20.65 | -3.54 | 88.33 | -16.51 | -3.56 | 88.32 | -17.65 | -3.3 | 88.57 | -14.14 | -1.95 | 89.92 | -5.78 | -1.3 | 90.59 | -2.81 | -0.57 | 91.3 | -7.68 | -1.1 | 90.78 | -6.89 | -0.44 | 91.43 |
| | SVHN | 2.46 | 3.06 | 68.59 | 6.05 | 5.04 | 70.58 | 2.43 | 2.57 | 68.03 | 5.7 | 4.64 | 70.18 | 7.89 | 3.3 | 68.84 | 6.98 | 3.7 | 69.12 | 6.02 | 5.32 | 70.67 | 8.24 | 4.68 | 70.21 |
| | CIFAR100 | 24.4 | 9.34 | 88.98 | 23.16 | 9.12 | 88.77 | 24.63 | 8.97 | 88.6 | 17.26 | 8.78 | 88.41 | 23.33 | 8.2 | 87.84 | 13.77 | 7.31 | 86.94 | 24.24 | 8.31 | 87.94 | 13.3 | 9.35 | 88.99 |
| DTD | Cars | -18.09 | -3.36 | 77.44 | -10.19 | -2.87 | 77.94 | -9.85 | -2.54 | 78.25 | -5.97 | -2.43 | 78.36 | -11.63 | -4.15 | 76.66 | -0.06 | 0.21 | 81.01 | -1.94 | 0.77 | 81.58 | -0.2 | 0.36 | 81.17 |
| | CIFAR10 | -36.94 | -6.55 | 88.37 | -13.28 | -4.32 | 90.59 | -13.4 | -4.74 | 90.17 | -9.33 | -1.81 | 93.11 | -12.64 | -4.99 | 89.92 | 0.57 | -0.44 | 94.48 | -0.76 | -0.22 | 94.7 | 0.4 | 0.21 | 95.12 |
| | CIFAR100 | -38.48 | -10.03 | 69.61 | -18.82 | -7.08 | 72.57 | -20.99 | -8.23 | 71.41 | -8.12 | -3.3 | 76.34 | -17.73 | -8.45 | 71.18 | 2.39 | 0.8 | 80.45 | -3.41 | -0.72 | 78.93 | 1.61 | 0.84 | 80.49 |
| | EuroSAT | -32.67 | -0.57 | 94.96 | -20.46 | -1.54 | 94.0 | -12.56 | -0.48 | 95.06 | -11.02 | -0.52 | 95.02 | -22.26 | -1.33 | 94.2 | 4.87 | 0.15 | 95.69 | -7.19 | 0.43 | 95.96 | 2.76 | 0.0 | 95.54 |
| | GTSRB | -18.13 | -7.02 | 79.69 | -7.89 | -3.52 | 83.15 | -13.89 | -2.21 | 84.51 | -4.41 | -1.61 | 85.01 | -5.76 | -2.62 | 83.97 | 0.34 | -0.39 | 86.18 | 2.11 | 0.69 | 87.32 | 1.43 | 1.3 | 87.93 |
| | MNIST | -24.64 | -0.95 | 97.75 | 3.59 | -0.51 | 98.16 | -7.74 | 0.02 | 98.35 | -5.74 | -0.32 | 98.34 | -7.92 | -0.14 | 98.49 | 5.03 | -0.02 | 98.69 | -2.84 | 0.17 | 98.86 | 2.6 | 0.15 | 98.84 |
| | RESISC45 | -34.51 | -5.0 | 86.87 | -11.0 | -2.52 | 89.37 | -13.37 | -2.65 | 89.22 | -10.52 | -1.48 | 90.4 | -19.27 | -4.48 | 87.38 | 0.33 | -0.13 | 91.75 | -1.86 | -0.38 | 91.49 | 1.21 | 0.14 | 92.02 |
| | SVHN | -17.89 | -5.9 | 59.64 | -8.6 | -5.14 | 60.18 | 0.31 | -3.28 | 62.14 | -3.49 | -2.61 | 63.03 | 0.55 | -3.54 | 61.9 | 1.84 | -0.11 | 65.3 | 7.88 | 0.76 | 66.31 | 3.81 | 1.23 | 66.69 |
| | DTD | 21.44 | 4.04 | 76.12 | 16.7 | 1.33 | 73.46 | 31.91 | 4.89 | 77.18 | 2.13 | 0.05 | 72.18 | 26.38 | 0.9 | 72.98 | 0.16 | 2.45 | 74.63 | 33.14 | 6.12 | 78.14 | 0.74 | 3.19 | 75.27 |
| EuroSAT | Cars | -18.44 | -7.0 | 73.8 | -14.04 | -8.34 | 72.45 | -12.34 | -7.85 | 72.95 | -13.0 | -6.27 | 74.53 | -4.89 | -2.79 | 78.01 | -3.95 | -1.84 | 78.97 | -1.37 | 0.42 | 81.22 | -1.08 | 0.53 | 81.33 |
| | CIFAR10 | -26.75 | -8.04 | 86.86 | -17.0 | -6.07 | 88.84 | -17.99 | -5.32 | 89.59 | -12.57 | -4.24 | 90.68 | -5.87 | -2.49 | 92.41 | -1.59 | -0.6 | 94.31 | -0.53 | 0.23 | 95.13 | 0.39 | 0.34 | 95.25 |
| | CIFAR100 | -35.66 | -11.37 | 68.26 | -23.68 | -8.87 | 70.76 | -24.64 | -5.78 | 69.85 | -17.79 | -6.03 | 73.59 | -6.43 | -4.0 | 75.62 | -0.83 | -0.56 | 79.08 | 1.23 | 0.46 | 80.09 | 2.27 | 1.06 | 80.7 |
| | DTD | -10.85 | -6.17 | 65.9 | -5.78 | -4.04 | 68.03 | -7.13 | -5.85 | 66.33 | -6.12 | -1.81 | 70.27 | -2.93 | -1.22 | 70.85 | -1.17 | 0.05 | 72.23 | 1.01 | 3.19 | 75.27 | 1.12 | 2.66 | 71.73 |
| | GTSRB | -9.49 | -8.53 | 78.14 | -5.78 | -7.54 | 79.16 | -6.39 | -8.39 | 78.35 | -4.29 | -4.35 | 82.31 | -1.27 | -4.05 | 82.6 | -0.58 | -1.2 | 85.53 | 4.56 | 0.94 | 87.67 | 4.51 | 1.98 | 88.6 |
| | MNIST | -6.85 | -1.2 | 97.45 | -6.92 | -1.35 | 97.32 | -7.76 | -1.32 | 97.29 | -5.24 | -1.48 | 97.14 | 3.21 | -0.54 | 98.16 | 0.04 | -0.2 | 98.44 | 5.54 | -0.07 | 98.57 | 2.82 | 0.13 | 98.74 |
| | RESISC45 | -24.64 | -3.89 | 87.97 | | -2.83 | 89.05 | -28.44 | -2.16 | 89.7 | -16.65 | -1.4 | 90.48 | -8.29 | -0.7 | 91.16 | -0.46 | 0.05 | 91.9 | -2.84 | 0.17 | 98.86 | 0.9 | 0.79 | 92.67 |
| | SVHN | -18.42 | -7.38 | 58.09 | -12.7 | -4.44 | 61.17 | -14.24 | -5.12 | 60.32 | -10.42 | -4.45 | 61.13 | 0.07 | -4.86 | 60.56 | 1.99 | -2.51 | 63.06 | 6.08 | 2.5 | 67.92 | 6.07 | 3.03 | 68.59 |
| | EuroSAT | 52.11 | 3.09 | 98.65 | 51.96 | 3.26 | 98.8 | 52.37 | 3.2 | 98.76 | 30.93 | 3.31 | 98.87 | 52.19 | 2.91 | 98.46 | 28.04 | 2.67 | 98.2 | 52.15 | 2.98 | 98.54 | 17.02 | 2.67 | 98.22 |
| GTSRB | Cars | -32.45 | -13.23 | 67.57 | -19.66 | -9.45 | 71.36 | -20.69 | -8.39 | 72.42 | -18.07 | -7.61 | 73.2 | -5.77 | -2.75 | 78.05 | -1.24 | -0.25 | 80.55 | -2.66 | 0.31 | 80.94 | -0.88 | 1.82 | 81.11 |
| | CIFAR10 | -61.3 | -17.99 | 76.92 | -31.36 | -8.35 | 86.57 | -41.57 | -12.51 | 82.4 | -15.19 | -4.71 | 90.21 | -40.89 | -11.41 | 83.51 | -1.77 | -0.2 | 94.72 | -1.71 | -0.38 | 94.53 | -0.14 | 0.0 | 94.92 |
| | CIFAR100 | -55.13 | -24.28 | 55.36 | -30.12 | -12.83 | 66.83 | -42.57 | -19.81 | 59.82 | -21.7 | -6.2 | 73.43 | -41.4 | -17.66 | 61.97 | 0.46 | 0.39 | 80.01 | -2.16 | -0.91 | 78.73 | 1.81 | 1.21 | 80.85 |
| | DTD | -24.68 | -14.63 | 57.45 | -10.0 | -7.5 | 64.57 | -17.18 | -9.2 | 62.87 | -10.74 | -4.35 | 68.35 | -6.6 | -2.13 | 69.64 | -1.54 | 1.76 | 73.72 | -2.55 | 1.65 | 73.62 | -0.32 | 1.17 | 73.03 |
| | EuroSAT | -36.94 | -5.0 | 90.56 | -19.31 | -2.91 | 92.63 | -21.91 | -4.72 | 90.81 | -17.7 | -1.11 | 94.43 | -16.5 | -3.31 | 92.22 | 3.94 | 0.0 | 95.54 | 1.17 | -0.43 | 95.11 | 2.57 | -0.06 | 95.48 |
| | MNIST | -6.53 | -0.81 | 97.89 | -6.43 | -0.25 | 98.35 | -13.95 | -0.63 | 98.02 | -1.72 | -0.09 | 98.6 | -4.72 | -0.39 | 98.33 | 3.33 | -0.06 | 98.58 | -2.23 | 0.13 | 98.8 | 6.78 | 0.02 | 98.66 |
| | RESISC45 | -46.86 | -10.35 | 81.52 | -19.24 | -5.44 | 86.43 | -35.92 | -8.73 | 83.14 | -18.17 | -2.43 | 89.44 | -13.68 | -2.92 | 88.95 | 0.1 | -0.02 | 91.86 | -2.6 | -0.48 | 91.4 | 0.32 | 0.11 | 91.96 |
| | SVHN | 2.22 | 18.35 | 83.82 | 5.07 | 16.66 | 82.25 | 2.74 | 16.81 | 82.08 | 10.11 | 18.12 | 83.65 | 15.22 | 16.15 | 81.75 | 9.68 | 7.79 | 73.39 | 10.08 | 8.67 | 74.17 | 7.94 | 5.9 | 71.45 |
| | GTSRB | 61.47 | 12.64 | 99.26 | 58.8 | 12.4 | 99.0 | 66.1 | 11.96 | 98.52 | 18.42 | 11.9 | 98.53 | 64.74 | 10.8 | 97.4 | 9.19 | 8.27 | 95.0 | 66.27 | 12.03 | 98.45 | 19.33 | 11.2 | 97.81 |
| MNIST | Cars | -29.96 | -17.16 | 63.64 | -29.3 | -16.44 | 64.38 | -29.55 | -18.58 | 62.24 | -44.11 | -15.91 | 64.02 | -4.56 | -2.18 | 78.65 | -2.47 | -0.41 | 80.38 | -3.89 | -0.9 | 79.93 | -3.25 | -0.08 | 80.23 |
| | CIFAR10 | -59.44 | -15.91 | 79.0 | -39.13 | -13.25 | 81.66 | -58.04 | -15.62 | 79.29 | -50.68 | -10.78 | 84.14 | -15.31 | -5.61 | 89.3 | -1.19 | 0.32 | 95.23 | -1.49 | -0.93 | 93.98 | -3.11 | -0.26 | 94.66 |
| | CIFAR100 | -56.3 | -22.92 | 56.72 | -42.41 | -18.2 | 61.44 | -50.72 | -21.05 | 58.57 | -55.09 | -16.71 | 62.94 | -23.77 | -9.44 | 70.19 | -5.18 | 0.11 | 79.76 | -5.64 | -1.26 | 78.37 | -4.43 | 0.46 | 80.11 |
| | DTD | -28.03 | -7.5 | 64.41 | -15.21 | -10.59 | 61.49 | -21.91 | -9.95 | 62.13 | -35.59 | -11.81 | 60.37 | -3.14 | -0.32 | 71.7 | -3.3 | 1.55 | 73.57 | -4.95 | 0.11 | 72.18 | -0.37 | 1.6 | 73.62 |
| | EuroSAT | -32.02 | -3.09 | 92.44 | -34.48 | -3.44 | 92.09 | -27.09 | -2.98 | 92.56 | -34.72 | -1.94 | 93.59 | -2.46 | -0.72 | 94.81 | -4.15 | -0.17 | 95.37 | -2.46 | 0.3 | 95.83 | -12.63 | 0.02 | 95.56 |
| | GTSRB | -25.71 | -6.02 | 80.59 | -15.46 | -4.22 | 82.45 | -20.58 | -6.25 | 80.39 | -23.1 | -0.02 | 86.63 | -5.57 | -2.45 | 84.19 | 1.26 | 2.77 | 89.33 | -0.91 | 1.85 | 88.47 | 3.71 | 2.89 | 89.49 |
| | RESISC45 | -42.9 | -6.68 | 85.19 | -39.11 | -7.29 | 84.59 | -37.94 | -7.05 | 84.84 | -51.0 | -5.94 | 85.94 | -5.46 | -1.56 | 90.32 | -1.68 | 0.24 | 92.11 | -4.68 | -0.06 | 91.81 | -3.19 | 0.26 | 92.13 |
| | SVHN | 17.26 | 10.15 | 75.71 | 16.11 | 13.21 | 78.82 | 15.42 | 11.42 | 76.94 | 13.47 | 14.99 | 80.57 | 12.9 | 6.06 | 71.45 | 12.41 | 7.51 | 72.85 | 22.94 | 15.31 | 80.84 | 19.95 | 16.76 | 82.31 |
| | MNIST | 51.19 | 0.88 | 99.62 | 50.56 | 0.98 | 99.63 | 51.28 | 1.02 | 99.67 | 21.34 | 0.98 | 99.68 | 51.02 | 0.8 | 99.43 | 5.51 | 0.5 | 99.18 | 51.18 | 0.98 | 99.6 | 8.35 | 0.83 | 99.52 |
| RESISC45 | Cars | -17.17 | -5.3 | 75.51 | -10.88 | -6.02 | 74.78 | -12.42 | -4.5 | 76.3 | -11.42 | -6.12 | 74.7 | -4.32 | -0.21 | 80.6 | -0.08 | 0.49 | 81.28 | -1.17 | 0.56 | 81.37 | 0.26 | 0.66 | 81.47 |
| | CIFAR10 | -33.88 | -8.43 | 86.49 | -15.74 | -5.1 | 89.61 | -18.47 | -5.84 | 89.08 | -10.76 | -5.24 | 91.47 | -7.58 | -2.04 | 92.88 | -0.89 | 0.0 | 94.91 | -0.6 | -0.2 | 93.92 | 0.13 | 0.13 | 95.04 |
| | CIFAR100 | -35.9 | -11.43 | 68.19 | -17.78 | -6.75 | 72.89 | -23.07 | -8.85 | 70.78 | -12.54 | -5.25 | 74.38 | -7.25 | -3.25 | 76.39 | 1.55 | 0.97 | 80.19 | 0.37 | -0.43 | 79.2 | 1.57 | 0.63 | 80.26 |
| | DTD | -12.66 | -4.47 | 67.77 | -6.44 | -2.39 | 69.52 | -8.56 | -3.24 | 68.83 | -3.51 | -0.48 | 71.6 | -3.4 | -0.74 | 71.33 | -0.59 | 2.07 | 74.15 | -1.81 | 1.01 | 73.4 | -0.43 | 2.98 | 74.79 |
| | EuroSAT | 2.5 | 1.06 | 96.61 | 8.85 | 1.07 | 96.61 | 5.22 | 0.94 | 96.28 | 9.15 | 1.07 | 96.61 | 12.3 | 1.13 | 96.67 | 13.11 | 1.26 | 96.8 | 13.74 | 0.91 | 96.44 | 12.76 | 0.76 | 96.3 |
| | GTSRB | -9.58 | -6.32 | 80.42 | -6.98 | -3.82 | 82.76 | -11.79 | -4.01 | 82.62 | -4.41 | -1.08 | 85.51 | -1.76 | -1.06 | 85.66 | 0.1 | 0.34 | 86.95 | -2.09 | 0.81 | 87.51 | 1.96 | 0.85 | 87.55 |
| | MNIST | -15.45 | -0.85 | 97.75 | -7.76 | -0.37 | 98.39 | -8.0 | -0.07 | 98.57 | -0.94 | -0.23 | 98.44 | 2.03 | -0.19 | 98.49 | 2.19 | -0.12 | 98.58 | 4.04 | -0.35 | 98.71 | 5.47 | -0.1 | 98.6 |
| | SVHN | -21.2 | -7.58 | 57.82 | -11.01 | -3.14 | 62.38 | -8.6 | -4.57 | 60.93 | -4.06 | -2.6 | 62.88 | 3.55 | -0.88 | 64.61 | 4.9 | 0.68 | 66.23 | 4.16 | 1.04 | 66.36 | 2.9 | 1.27 | 66.89 |
| | RESISC45 | 33.49 | 3.98 | 95.84 | 30.4 | 3.94 | 95.79 | 35.1 | 3.9 | 95.76 | 13.65 | 3.68 | 95.56 | 33.14 | 2.19 | 94.05 | 8.14 | 2.25 | 94.13 | 34.76 | 3.37 | 95.22 | 10.92 | 3.25 | 95.13 |
| SVHN | Cars | -39.37 | -17.56 | 63.23 | -31.22 | -17.22 | 63.59 | -33.06 | -18.38 | 62.43 | -30.44 | -15.28 | 65.51 | -5.66 | -2.92 | 77.9 | -6.63 | -2.0 | 78.8 | -6.5 | -2.04 | 78.76 | -8.73 | -2.89 | 77.91 |
| | CIFAR10 | -44.46 | -13.57 | 81.34 | -39.98 | -13.84 | 81.06 | -42.24 | -14.41 | 80.5 | -32.49 | -11.44 | 83.49 | -21.11 | -7.47 | 87.45 | -24.95 | -5.66 | 89.25 | -7.25 | -2.16 | 92.75 | -6.83 | -1.86 | 93.06 |
| | CIFAR100 | -52.73 | -21.64 | 58.0 | -43.38 | -22.93 | 56.72 | -46.16 | -24.14 | 55.51 | -41.28 | -18.07 | 61.59 | -28.52 | -11.84 | 67.8 | -30.95 | -10.09 | 69.56 | -12.88 | -5.26 | 74.38 | -11.03 | -2.73 | 76.9 |
| | DTD | -28.51 | -19.2 | 52.87 | -20.11 | -15.48 | 56.6 | -24.1 | -18.09 | 53.88 | -20.48 | -15.11 | 56.97 | -5.8 | -1.06 | 71.17 | -7.45 | -0.9 | 71.12 | -4.47 | -0.48 | 71.76 | -1.86 | 0.31 | 72.29 |
| | EuroSAT | -36.17 | -4.31 | 91.02 | -25.59 | -4.3 | 91.24 | -34.87 | -5.7 | 89.83 | -36.09 | -6.04 | 89.5 | -19.98 | -2.26 | 93.28 | -27.78 | -2.56 | 92.98 | -2.61 | -0.78 | 94.76 | -13.74 | -0.78 | 94.76 |
| | GTSRB | -21.91 | 0.0 | 86.68 | -10.66 | 0.95 | 87.72 | -8.27 | 1.78 | 88.41 | -9.62 | 4.95 | 91.57 | 1.8 | 5.04 | 91.7 | -0.06 | 6.03 | 92.64 | 5.49 | 6.03 | 92.66 | 6.45 | 6.23 | 92.9 |
| | MNIST | 36.16 | -0.04 | 98.61 | 30.61 | -0.19 | 98.47 | 38.35 | 0.0 | 98.63 | 7.66 | 0.15 | 98.81 | 31.19 | 0.23 | 98.91 | 11.39 | -0.06 | 98.61 | 41.91 | 0.66 | 99.15 | 19.55 | 0.49 | 99.06 |
| | RESISC45 | -44.29 | -9.59 | 82.29 | -34.94 | -10.1 | 81.78 | -37.16 | -10.0 | 81.87 | -40.17 | -8.97 | 82.9 | -8.44 | -1.95 | 89.92 | -7.84 | -1.65 | 90.22 | -4.56 | -1.38 | 90.49 | -5.56 | -1.0 | 90.87 |
| | SVHN | 65.0 | 31.92 | 97.44 | 64.38 | 31.86 | 97.4 | 65.58 | 31.79 | 97.3 | 51.28 | 31.98 | 97.5 | 64.84 | 31.02 | 96.5 | 53.39 | 31.04 | 96.54 | 65.25 | 31.36 | 96.97 | 50.92 | 31.38 | 96.95 |

Table 9: $\Delta_{ZS}$, $\Delta_{LP}$ and $\mathcal{A}_{LP}$ for models fully fine-tuned on 9 different classification datasets. LP-init-LDIFS outperforms other fine-tuning baselines and even LP-init-L2SP in minimizing concept forgetting, while also outperforming LP-init-L2SP on the fine-tuned task performance.

