# OpenReview forum: "Fine-tuning can cripple your foundation model; preserving features may be the solution"
_TMLR — Accepted by TMLR_

### Review · Reviewer_wCd4 · 2024-03-04

**Summary Of Contributions:**

The paper shows a concept forgetting phenomenon in the fine-tuning process of a foundation model, where fine-tuning on one task leads to performance deterioration on other tasks. The paper presents extensive experiments to demonstrate the concept forgetting issue on 9 datasets. A feature preserving regularization (LDIFS) is proposed to fix this issue and shown to be effective on multiple datasets compared with several baselines.

**Audience:**

Yes

**Broader Impact Concerns:**

Not applicable.

**Claims And Evidence:**

Yes

**Requested Changes:**

1. Add a larger foundation model in the empirical study

2. Explore the reason why LDIFS is better than parameter constraint methods.

3. Explain the effect of training sample size in a downstream task to the effectiveness of the proposed approach.

**Strengths And Weaknesses:**

Strengths:

The experiment is comprehensive in terms of downstream tasks.


Weaknesses:

1. The paper only considers one foundation model, i.e., CLIP ViT-B/32. It is unclear whether the conclusion generalizes to larger models like CLIP ViT-L.

2. The reason why preserving feature space is better than preserving parameter space is not well explored.

---

> ### Author Response · Authors · 2024-03-25
> **Model ablations, LDIFS vs parameter constraint methods, Effect of training sample size**
>
> Thank you for taking the time to review our work and for your feedback! We provide our responses below.
>
> **”The paper only considers one foundation model, i.e., CLIP ViT-B/32. It is unclear whether the conclusion generalizes to larger models like CLIP ViT-L.”**
>
> **”Add a larger foundation model in the empirical study”**
>
> * We have performed an extensive ablation across both different available pre-trained architectures of CLIP as well as other image-text foundation models like FLAVA and vision only models like DINOv2 and MAE, on fine-tuning on ImageNet.
> * For CLIP, DINOv2 and MAE, we also ablated across model sizes using both ViT-B and ViT-L architectures. In particular, for CLIP, we have ResNet-50, ViT-B/32, ViT-B/16 and ViT-L/14 architectures.
> * Please see our general response “Concept forgetting beyond CLIP ViT-B/32” for details.
>
> Our observations on performance across model size is as follows.
> * We found concept forgetting to exist across all model sizes in CLIP. This is indicated through consistent negative $\Delta_{\mathrm{LP}}$ values for all fine-tuning baselines except LDIFS.
> * Comparing between model sizes, we found larger models to have progressively lower levels of concept forgetting. This is not just true for CLIP but is also evident from results on other models like DINOv2 and MAE.
> * Our proposed baseline LP-init-LDIFS consistently outperformed other competitive fine-tuning baselines on all these models across all model sizes.
>
> **”Explore the reason why LDIFS is better than parameter constraint methods.”**
>
> * Neural networks are well-known to be overparameterized where multiple parameter or weight configurations which can be relatively distant, can encode similar model behaviour.
> * As a toy example, consider the following two neurons: $f_1(x) = ReLU(w_1.x + b_1)$ and $f_2(x) = ReLU(w_2.x + b_2)$ where $w_1 = -100, b_1 = -100, w_2 = -1, b_2 = -1$ and $0 \leq x \leq 1$. We can see that $\forall x$, $f_1(x) = f_2(x) = 0$, indicating that the behaviour of these two neurons is always the same even though the parameter space distance between these two neurons is significantly large.
> * On the other hand if $w_1 = -1$ and $b_1 = 1$, $f_1(x) = (1-x) \neq f_2(x)$ even though the weight space distance between $f_1$ and $f_2$ is much lower.
> * Thus, while parameter or weight space distance can indeed be regularized, in order to capture and preserve model behaviour, regularizing the feature space or activation space distance is a better option.
> * This is further evident from our empirical observations in Fig 6 and 7 of the revised draft. LDIFS while having a larger parameter space distance from the pre-trained model still has near 0 feature space distance, thereby preserving the model behaviour while also attaining competitive downstream task performance, something that L2SP, using its parameter space regularizer, fails to achieve.
>
> **"Explain the effect of training sample size in a downstream task to the effectiveness of the proposed approach."**
>
> * The 10 datasets we use for the results in Table 1 of the main paper has a high variance in the number of training samples.
> * Specifically, the number of training samples ranges from 1880 samples in DTD to 1.28 M samples in ImageNet.
> * As evident from Table 1, across all these different training sample sizes, LP-init-LDIFS remains performant and is able to lower concept forgetting.
> * As such we don't observe any other relation between training sample size and the amount of concept forgetting as quantified by $\Delta_{\mathrm{LP}}$.

---

### Review · Reviewer_RQ9w · 2024-03-06

**Summary Of Contributions:**

I summarize the contributions of this paper as follows:
1. The paper addresses a particularly practical issue in an era where foundation models pretrained on massive-scale data are widely used, that is, concept forgetting.
2. The authors formally define the concept forgetting and empirically show that existing end-to-end finetuning methods except L2SP severely suffers from the concept forgetting.
3. The paper proposes a simple yet effective regularizer, LDIFS, to alleviate the concept forgetting.

**Audience:**

Yes

**Claims And Evidence:**

Yes

**Requested Changes:**

Overall, the authors conducted extensive experiments, which effectively highlight the contributions of the paper. Please address the weaknesses mentioned above.

Minor question: Does "LPIPS" in the provided code refer to "LDIFS"?

**Strengths And Weaknesses:**

Strengths
- Overall, the paper is well-written and easy to follow. Especially, Figure 1 and introducing the proposed method, LDIFS from the analyses of L2SP justify using linear probe (LP) for measuring concept forgetting and the validity of LDIFS, respectively.
- Extensive experiments help readers understand the newly defined problem, "concept forgetting", and validate the effectiveness of LDIFS.
- Enormous computational resources are utilized for experiments.

Weaknesses
- Editorial comments:
  - TMLR can merge the Appendix just after the main paper, which would enhance readability.
  - The analyses of LDIFS on parameter space and feature space distance, Figure 11 and Figure 12 in the Appendix, seem to be critical for deeply understanding the regularizer. Please move the analyses to the main paper, and also add discussion about why applying the regularizer on the target dataset also minimizes feature space distance, hence concept forgetting on other datasets (utilizing features from intermediate layers might be one reason).
- With regard to the above comment, further discussion is needed for validating the design choice made for LDIFS: The authors should provide not only the delta LP values for LL (using the last layer only) but also the actual feature space distance on non-target datasets to clarify that the improvement in concept forgetting comes from reduced feature space distance, as the authors intended.

---

> ### Author Response · Authors · 2024-03-25
> **Edits & Discussion around last layer features**
>
> Thank you for taking the time to review our work and for your feedback! We provide our responses below.
>
> **"The analyses of LDIFS on parameter space and feature space distance, Figure 11 and Figure 12 in the Appendix, seem to be critical for deeply understanding the regularizer. Please move the analyses to the main paper,"**
>
> We have moved this analysis to the main paper in the revised draft.
>
> **"and also add discussion about why applying the regularizer on the target dataset also minimizes feature space distance, hence concept forgetting on other datasets (utilizing features from intermediate layers might be one reason).
> With regard to the above comment, further discussion is needed for validating the design choice made for LDIFS: The authors should provide not only the delta LP values for LL (using the last layer only) but also the actual feature space distance on non-target datasets to clarify that the improvement in concept forgetting comes from reduced feature space distance, as the authors intended."**
>
> Thank you for raising this point.
>
> * We have already provided empirical evidence that shows that minimizing LDIFS on the target dataset also minimizes the feature distance on non-target datasets. This is through Fig 11 and 12 in the appendix (Fig 6 and 7 in the revised draft of the main paper).
> * Following your suggestion, to see the behaviour when applying LDIFS using just last layer features, we plot the feature space distance including this baseline in Section C.9, Fig 13 of the appendix. Indeed, when using the last layer ablation, the distance in feature space (using the full feature vector) is not minimized both on the target set as well as other datasets, which correlates with worse performance on $A_{\mathrm{LP}}$ for the LL baseline (Fig 10 main paper). We have added this discussion in Section C.9 of the appendix.
>
> **"Minor question: Does "LPIPS" in the provided code refer to "LDIFS"?"**
>
> It does indeed. Thanks for noticing. We will fix the naming before releasing the code.

---

### Review · Reviewer_2J6R · 2024-03-11

**Summary Of Contributions:**

This paper addresses a critical issue in the fine-tuning of pre-trained foundation models: "concept forgetting," where the fine-tuned model's ability to recognize diverse concepts diminishes. The authors observe that while fine-tuning on specific datasets improves performance on downstream tasks, it often reduces the model's pre-trained knowledge on other domain of concepts, a phenomenon they term "concept forgetting." To counteract this, they propose a new fine-tuning regularizer named LDIFS, which is designed to preserve pre-trained knowledge while acquiring new concepts. Extensive experiments across 10 fine-tuning tasks demonstrate LDIFS's effectiveness in reducing concept forgetting and its superior performance in continual fine-tuning compared to standard fine-tuning and continual learning baselines.

**Audience:**

Yes

**Broader Impact Concerns:**

It will be a plus if the authors can also apply the method in fairness settings.

**Claims And Evidence:**

No

**Requested Changes:**

Please check cons for details.

**Strengths And Weaknesses:**

Pros:
1. The proposed method to address the issue is straightforward and intuitive.
2. The paper is well presented and overall easy to follow.
3. The empirical analysis is interesting.

Cons:
1. The major concern of the reviewer lies in whether the identified problem and setting is really important in practice or not. In practice, when one plan to fine-tune a foundation model on a downstream dataset, the domain of the application and the concepts that are important are usually already defined. For example, if the downstream application requires ability on understanding both satellite images and remote sensing, one can easily improve the model's ability on these domains by using joint training data from these two domains. It is also questionable whether the proposed method can address possible problems in this practical cases where there could be certain conflicts in the desired sets of concepts in the application.

2. The current empirical analysis only covers one pretrained model. It is important to show that the method generalizes to different model architectures, and especially different scales of models in terms of both pretraining data and model size. Moreover, despite the popularity of CLIP, there are also vision only pretrained models such as MAE. It is also important to udnerstand whether the identified problem and the proposed method is generalizable enough to handle different pretraining methods.

---

> ### Author Response · Authors · 2024-03-25
> **Motivating problem setting & model ablations**
>
> Thank you for taking the time to review our work and for your feedback! We provide our responses below.
>
> **"The major concern of the reviewer lies in whether the identified problem and setting is really important in practice or not. In practice, when one plan to fine-tune a foundation model on a downstream dataset, the domain of the application and the concepts that are important are usually already defined. For example, if the downstream application requires ability on understanding both satellite images and remote sensing, one can easily improve the model's ability on these domains by using joint training data from these two domains. It is also questionable whether the proposed method can address possible problems in this practical cases where there could be certain conflicts in the desired sets of concepts in the application."**
>
> Thank you for raising this point.
>
> ***Use case for fine-tuning***
> * We agree that the most conventional or common use case for fine-tuning is to get state-of-the-art result on a downstream task or domain. That is generally what most fine-tuning methods are designed to optimize on.
> * However, we argue that that is *not the only use case for fine-tuning, especially for a foundation model*.
> * The pre-training procedure for a foundation model is particularly expensive. Yet, there is ample evidence that foundation models have knowledge gaps, especially in domains which are not well represented in their pre-training sets.
> * Two well-known examples are CLIP’s underperformance on satellite imagery and medical domains, and knowledge cut-offs for LLMs like ChatGPT. Furthermore, in domains like medicine, there is a constant influx of new knowledge in the form of new diseases, new treatments, new procedures etc.
> * Thus, there is a clear necessity for “updating” foundation models on their knowledge gaps without having to retrain them from scratch.
> * This requires fine-tuning while also preserving the previously trained knowledge base in a foundation model. Therefore, we argue in our work that fine-tuning methods should also take into account concept forgetting as a phenomenon that should be minimized, especially for foundation models.
> * Finally, even for a user who cares only about downstream task performance, we provide the option to choose a fine-tuning method which attains close to SOTA on the downstream task while also preserving the knowledge base of the model. The user is then free to choose.
>
> ***Joint training data from multiple domains***
> * Empirically, we already show through our results on ImageNet that LDIFS can be effective even when joint training on multiple domains.
> * ImageNet contains data from different sub-domains like animals, vehicles, food etc. In Table 1 of the main paper, we show that LP-init-LDIFS can produce very competitive performance on ImageNet while also producing a positive forward transfer on other datasets.
>
> ***Conflicts in domains***
> * Thank you for this interesting point.
> * It is true that we cannot guarantee LDIFS’s performance when there is a conflict in the domains that the model is trained on. However, no fine-tuning method can guarantee performance in this situation.
> * As a simple example, consider jointly training on two tasks where one task is to classify right arrows from left arrows and another task specifically requires rotation invariance. In this situation, there can’t be any performance guarantee from any fine-tuning method simply due to the nature of the tasks themselves.
>
>
> **"The current empirical analysis only covers one pretrained model. It is important to show that the method generalizes to different model architectures, and especially different scales of models in terms of both pretraining data and model size. Moreover, despite the popularity of CLIP, there are also vision only pretrained models such as MAE. It is also important to udnerstand whether the identified problem and the proposed method is generalizable enough to handle different pretraining methods."**
>
> Thank you for this comment.
> * We have performed an extensive ablation across both different available pre-trained architectures of CLIP as well as other image-text foundation models like FLAVA and vision only models like DINOv2 and MAE, on fine-tuning on ImageNet.
> * For CLIP, DINOv2 and MAE, we have also ablated across different model sizes like ViT-B and ViT-L.
> * For details, please see our general response: “Concept forgetting beyond CLIP ViT-B/32”.
> * We found concept forgetting to exist across model types (CLIP, FLAVA, DINOv2 and MAE) as well as architectures (ViT-B, ViT-L).
> * We also found our proposed baseline LP-init-LDIFS to consistently outperform other competitive fine-tuning baselines across all model types and architectures.

---

### Review · Reviewer_FFde · 2024-03-11

**Summary Of Contributions:**

The paper studied the "concept forgetting" phenomena in finetuned foundation models. After trained on a specific task, the finetuned models often lost their ability to recoganize general-domain concepts. To tackle the concept forgetting problem, the author propose to regularize the finetuning process by minimizing the L2 distance between features from the original model and the finetuned model (called LDIFS in the paper).

In the experiments, the author showed that the concept forgetting phenomena, which is measured by difference of linear probing accuracy, has a strong correlation with feature-space distance. The proposed LDIFS regularizer is effective in addressing the concept forgetting problem.

**Audience:**

Yes

**Claims And Evidence:**

Yes

**Requested Changes:**

Comparison with other continual learning algorithms
Comparison with CLIP in different sizes to see how scale up the model size will impact concept forgetting.

**Strengths And Weaknesses:**

Pros:

1. The LDIFS idea is intuitive and effective in addressing the concept drifting problem.
2. The author conducted experiments on a diverse collection of datasets.


Cons:

1. LDIFS is a type of continual learning algorithm but author has not compared with a well-estabilished baseline called continual learning with rehearsal. The idea of continual learning with rehearsal is to finetune the model on the combination of down-stream samples and a subset of general-domain samples (called rehearsal memory). It has been shown to be beneficial for finetuning instruction-tuned language models in [Fine-tuned Language Models are Continual Learners](https://aclanthology.org/2022.emnlp-main.410.pdf).In fact, how to continually adapting a foundation model has been studied in other papers such as [Lifelong Pretraining: Continually Adapting Language Models to Emerging Corpora](https://aclanthology.org/2022.naacl-main.351.pdf). The LDIFS algorithm is relevant to previous works that apply knowledge distillation in the feature space for continual adaptation.

2. The author only compared different algorithm with impact of continual finetuning with ViT-B/32 version of CLIP. It is unclear if the same conclusion will transfer to models with different sizes. In fact, the transfer learning capability of large-scale models is relevant to the size (see [Scaling Laws for Transfer
](https://arxiv.org/abs/2102.01293) ). It is unclear if the concept forgetting problem will be alleviated or worsen for larger-scale CLIP models.

---

> ### Author Response · Authors · 2024-03-25
> **Continual Learning with Rehearsal and Model Ablations**
>
> Thank you for taking the time to review our work and for your feedback! We provide our responses below.
>
> **"LDIFS is a type of continual learning algorithm but author has not compared with a well-estabilished baseline called continual learning with rehearsal. The idea of continual learning with rehearsal is to finetune the model on the combination of down-stream samples and a subset of general-domain samples (called rehearsal memory). It has been shown to be beneficial for finetuning instruction-tuned language models in Fine-tuned Language Models are Continual Learners."**
>
> **"Comparison with other continual learning algorithms"**
>
> Thank you for this point.
>
> ***On comparing with continual learning baselines using rehearsal***
> * In the main paper, we already compare LP-init-LDIFS with 5 well-known continual learning baselines: LwF, LFL, iCARL, Distillation + Retrospection (D+R) and ZSCL.
> * This is in Table 3 of the main paper and we find LP-init-LDIFS to consistently outperform all other baselines on all 3 sequence tasks.
> * Among these baselines, iCARL, D+R and ZSCL all use exemplar, retrospection or reference samples which are a form of rehearsal from previously seen tasks or general domain samples.
>
> ***On LDIFS with rehearsal***
> * We kept LDIFS rehearsal free, as apriori, from a pre-trained foundation model, we are assuming that we may not know what we would like to preserve from the pre-trained model, as we fine-tune the model on a new task.
> * In addition, the pre-training data for foundation models is often unavailable or difficult to parse, so getting domain specific exemplar samples to preserve from the pre-training set is not simple.
> * Since we may not know what we want to preserve when fine-tuning, ideally, we want to preserve *everything*, i.e., all the pre-trained knowledge and this is precisely our motivation when designing LDIFS. Hence, we kept it rehearsal free from domain specific samples.
>
> ***Ablation with joint training***
> * In addition to the 5 baselines we compare with in Table 3, we now have a joint-training baseline where we use the full training set of all previous tasks along with the current task in the sequence at any point.
> * Please see our general response “Comparing LP-init-LDIFS with joint training” for details and results.
>
> Our main observations from this experiment are as follows.
> * LP-init-LDIFS is competitive but slightly underperforms the joint training baseline on previous tasks in the sequence. Note that joint training sees the full training data of previous tasks as well.
> * LP-init-LDIFS is competitive with joint training on the current task in the sequence.
> * LP-init-LDIFS significantly outperforms joint training on preserving pre-trained performance on tasks which aren’t part of the sequence and we don’t fine-tune on.
>
>
> **"The author only compared different algorithm with impact of continual finetuning with ViT-B/32 version of CLIP. It is unclear if the same conclusion will transfer to models with different sizes. In fact, the transfer learning capability of large-scale models is relevant to the size (see Scaling Laws for Transfer ). It is unclear if the concept forgetting problem will be alleviated or worsen for larger-scale CLIP models."**
>
> Thank you for this comment.
> * We have performed an extensive ablation across both different available pre-trained architectures of CLIP as well as other image-text foundation models like FLAVA and vision only models like DINOv2 and MAE, on fine-tuning on ImageNet.
> * For CLIP, DINOv2 and MAE, we also ablated across model sizes using both ViT-B and ViT-L architectures. In particular, for CLIP, we have ResNet-50, ViT-B/32, ViT-B/16 and ViT-L/14 architectures.
> * Please see our general response “Concept forgetting beyond CLIP ViT-B/32” for details.
>
> Our observations on performance across model size is as follows.
> * We found concept forgetting to exist across all model sizes in CLIP. This is indicated through consistent negative $\Delta_{\mathrm{LP}}$ values for all fine-tuning baselines except LDIFS.
> * Comparing between model sizes, we found larger models to have progressively lower levels of concept forgetting. This is not just true for CLIP but is also evident from results on other models like DINOv2 and MAE.
> * Our proposed baseline LP-init-LDIFS consistently outperformed other competitive fine-tuning baselines on all these models across all model sizes.
>
> **"In fact, how to continually adapting a foundation model has been studied in other papers such as Lifelong Pretraining: Continually Adapting Language Models to Emerging Corpora. The LDIFS algorithm is relevant to previous works that apply knowledge distillation in the feature space for continual adaptation."**
>
> * Thank you. We will include this and other relevant citations in our related works section.
> * In our current Related works (Section 6), we have thoroughly discussed the similarities and differences of distillation based methods from LDIFS.

---

### Review · Reviewer_UAdC · 2024-03-11

**Summary Of Contributions:**

The paper investigates the phenomenon of "concept forgetting" during fine-tuning of foundation models especially CLIP, where the fine-tuned model loses its ability to recognise concepts outside of the downstream task it was fine-tuned on. It proposed a new regularizer called LDIFS (l2 distance in feature space) that minimises distance between the original and fine-tuned models in feature space that seems to tackle the forgetting problem effectively.

**Audience:**

Yes

**Claims And Evidence:**

Yes

**Requested Changes:**

Please refer to the weaknesses above. Especially the last three points are related to this.

**Strengths And Weaknesses:**

Strengths:
 - The paper addresses an important problem of “concept forgetting” of foundation models.
 - The paper is well written and easy to understand

Weaknesses:
 - **Novelty of Concept Forgetting**: The concept forgetting phenomenon seems very similar to catastrophic forgetting. Although the authors have tried to differentiate the two terms, they both seem the same to me. The authors define concept forgetting as  the drop in performance on a dataset different from the pre-trained dataset in foundation models. However, this also falls under the purview of catastrophic forgetting and has been explored by earlier works ([1], [2], [3], etc). Hence, I am not convinced that concept forgetting is a new phenomenon. Additionally, the authors need to compare their results with these approaches as well.
 - **Section 3.1: Subsection - Catastrophic forgetting vs concept forgetting**: First of all, it is very confusing when it is told that in the pre-foundation model era, the pre-training tasks were fully supervised. Yes its true. But this is also true for the so called foundation model CLIP that the authors have used as the to-go model. Going by the traditional definition of full supervision i.e., using input example and corresponding label/annotation, CLIP pretraining is also fully supervised as it uses paired image-caption data. The training uses the information of which caption is for which image and is not oblivious to it. So, taking away the factor of the 'use of supervision while pretraining', the subsection boils down to justifying the difference between catastrophic and concept forgetting as whether or not the pretraining dataset is small or big. Probably, this is a major source of confusion to some of the lines written later in the subsection - e.g., "it was possible to roughly quantify the degree of damage ... on a new downstream task". Why is it 'roughly' quantifiable? Then why is not 'trivial' for the foundation models. I can understand that the pre-training dataset is not often accessible. In absence of the pre-training dataset, the obvious choice would be to set aside a proxy dataset on which the effect of finetuning can be observed. And precisely this is done by the authors (the use of a target dataset). But why does having a target dataset in absence of the pretraining dataset need to rename catastrophic forgetting to concept forgetting? It is still forgetting and frankly speaking, if one does not have the original pretraining dataset, there is no way to tell precisely if the 'concepts' are forgotten or not. This is because, what all concepts the foundation model learnt in the first place can't be known or tested. So, in brief, why the term 'concept forgetting' is being differentiated with 'catastrophic forgetting' is not clear. c. Incremental Novelty: The paper proposes an approach similar to knowledge distillation which has been explored before. LwF [10 in main paper] and ICARL [11 in main paper] are two classical continual learning papers which have used such approaches. While the authors apply the feature regularisation over all layers as compared to final layers by knowledge distillation, the novelty seems incremental.
 - **More results: Section 5**: The authors showed comparisons on similar Continual Learning (CL) Datasets but not the same ones. Why not use the same datasets as is generally followed in CL literature - say 5-datasets or ImageNet-R or CUB etc. Additionally,  there exists prior works applying Continual Learning on CLIP models [4, 5, 6, 7 ]. The authors need to compare with them.
 - **Doubts on Claim**: The claim that full fine-tuning can cripple your foundation model is somewhat misleading as works like SLCA[8] and FSA[9] has proven otherwise. The authors need to show results with these approaches as well.

References:

1. Smith, James Seale, et al. "Coda-prompt: Continual decomposed attention-based prompting for rehearsal-free continual learning." Proceedings of the IEEE/CVF Conference on Computer Vision and Pattern Recognition. 2023.
2. Wang, Zifeng, et al. "Learning to prompt for continual learning." Proceedings of the IEEE/CVF Conference on Computer Vision and Pattern Recognition. 2022.
3. Wang, Zifeng, et al. "Dualprompt: Complementary prompting for rehearsal-free continual learning." European Conference on Computer Vision. Cham: Springer Nature Switzerland, 2022.
4. D'Alessandro, Marco, et al. "Multimodal Parameter-Efficient Few-Shot Class Incremental Learning." Proceedings of the IEEE/CVF International Conference on Computer Vision. 2023
5. Thengane, Vishal, et al. "Clip model is an efficient continual learner." arXiv preprint arXiv:2210.03114 (2022)
6. Chen, Haoran, et al. "Promptfusion: Decoupling stability and plasticity for continual learning." arXiv preprint arXiv:2303.07223 (2023)
7. Zhou, Da-Wei, et al. "Learning without forgetting for vision-language models." arXiv preprint arXiv:2305.19270 (2023)
8. Zhang, Gengwei, et al. "Slca: Slow learner with classifier alignment for continual learning on a pre-trained model." Proceedings of the IEEE/CVF International Conference on Computer Vision. 2023
9. Panos, Aristeidis, et al. "First session adaptation: A strong replay-free baseline for class-incremental learning." Proceedings of the IEEE/CVF International Conference on Computer Vision. 2023.
10. Li, Zhizhong, and Derek Hoiem. "Learning without forgetting." IEEE transactions on pattern analysis and machine intelligence 40.12 (2017): 2935-2947
11. Rebuffi, Sylvestre-Alvise, et al. "icarl: Incremental classifier and representation learning." Proceedings of the IEEE conference on Computer Vision and Pattern Recognition. 2017
12. Radford, Alec, et al. "Learning transferable visual models from natural language supervision." International conference on machine learning. PMLR, 2021

---

> ### Author Response · Authors · 2024-03-25
> **Novelty of Concept Forgetting**
>
> Thank you for taking the time to review our work and for your feedback. We provide responses below.
>
>
> **"Novelty of Concept Forgetting: The concept forgetting phenomenon seems very similar to catastrophic forgetting. Although the authors have tried to differentiate the two terms, they both seem the same to me. The authors define concept forgetting as the drop in performance on a dataset different from the pre-trained dataset in foundation models."**
> * We don’t define concept forgetting as the “drop in performance on a dataset different from the pre-trained dataset in foundation models”. In fact, for foundation models, we don’t have access to pre-trained datasets.
> * We define it as the forgetting of concepts which were present in the pre-trained dataset, caused by fine-tuning on a downstream task. The forgotten concepts are not present in the downstream task and hence, even though the pre-trained model could recognize those concepts, it is less able to do so, once it has been fine-tuned.
>
> **"However, this also falls under the purview of catastrophic forgetting and has been explored by earlier works ([1], [2], [3], etc). …Additionally, the authors need to compare their results with these approaches as well."**
>
> * Coda-Prompt [1], L2P [2] and DualPrompt [3] are continual learning approaches which leverage the pre-training of a ViT-B/16 on ImageNet, to design prompt based methods which lead to improvements over previous rehearsal based continual learning baselines.
> * The motivation and the problem that these works are trying to solve are different to our work. These papers are trying to leverage pre-training as a means to improve on classic continual learning approaches which had additional constraints like rehearsal.
> * Therefore, they evaluate and benchmark their approaches on well-known class-incremental learning datasets like Split-CIFAR100 or Split-ImageNet-R.
> * In this work, we are not aiming to leverage the pre-trained encoders of foundation models to improve performance on well-known existing continual learning benchmarks.
> * Rather, we are motivated to design a fine-tuning approach specifically to improve pre-trained foundation models on a potentially new downstream task while preserving their vast pre-trained knowledge.
> * Note that in the paper, we don’t start our analysis comparing with classic continual learning approaches, rather we start looking at state-of-the-art fine-tuning approaches for foundation models, identify problems related to forgetting in these approaches, and work backwards to design a fine-tuning solution that also minimizes concept forgetting.
> * Finally, we have compared LP-init-LDIFS with [1,2,3] on Split-ImageNet-R in a separate response here.
>
>  **"Hence, I am not convinced that concept forgetting is a new phenomenon."**
>
> * We are not claiming in our work that the underlying phenomenon causing concept forgetting is different from the one causing catastrophic forgetting.

---

> ### Author Response · Authors · 2024-03-25
> **Catastrophic Forgetting vs Concept Forgetting**
>
> **"First of all, it is very confusing when it is told that in the pre-foundation model era, the pre-training tasks were fully supervised. Yes its true. But this is also true for the so called foundation model CLIP that the authors have used as the to-go model. Going by the traditional definition of full supervision i.e., using input example and corresponding label/annotation, CLIP pretraining is also fully supervised as it uses paired image-caption data. The training uses the information of which caption is for which image and is not oblivious to it."**
>
> * We believe CLIP pre-training to lie within the regime of weakly supervised pre-training as opposed to fully supervised ones.
> * You are right in saying that CLIP is trained on pairs of images and captions. However, there are distinct differences between CLIP pre-training and fully supervised training.
> * First, the pre-training loss is contrastive loss, a well-known self-supervised loss function, as opposed to cross-entropy, the go-to loss function for supervised classification training.
> * Second, CLIP is not trained on explicit labels or annotations of concepts in a structured format i.e., it is not trained on a fixed number of labels or classes. Rather the paired data in CLIP is used to train an alignment between two modalities where the text description is unstructured.
> * Third, the purpose behind CLIP pre-training is similar to self-supervised training which is to learn suitable features applicable to a wide variety of downstream tasks, rather than supervised training which is to maximize performance on a single downstream task.
> * Thus, by pre-trained foundation models, we mean models which have been trained in a self-supervised manner on a large (several million or billion scale) pre-training dataset. Since the advent of such models, state-of-the-art in machine learning tasks has shifted from fully supervised models specifically trained on those tasks to models which are obtained either directly or through relatively little adaptation from a pre-trained foundation model. This paradigm change is what we mean by the foundation model era.
>
> **"So, taking away the factor of the 'use of supervision while pretraining', the subsection boils down to justifying the difference between catastrophic and concept forgetting as whether or not the pretraining dataset is small or big. Probably, this is a major source of confusion to some of the lines written later in the subsection - e.g., "it was possible to roughly quantify the degree of damage ... on a new downstream task". Why is it 'roughly' quantifiable?"**
>
> * In a fully supervised setting like classification,, we measure test set accuracy with the i.i.d assumption that the test samples are drawn from the same data distribution as the training samples.
> * However, since the test set is limited and finite, test set performance is only a proxy to performance on the data distribution. A drop in test set accuracy is a proxy to the drop in performance on the data distribution. Hence, the use of the phrasing “roughly quantify the degree of damage”.
>
> **"Then why is not 'trivial' for the foundation models."**
>
> * In the corresponding line from the paper, we are mentioning that unlike a fully supervised model, like a classification model, where we know the exhaustive set of concepts which the model has been trained on, in a pre-trained foundation model, we don’t have that knowledge due to the inaccessibility of the pre-training dataset.
> * Hence, it is not trivial to quantify what a pre-trained foundation model knows.

---

> ### Author Response · Authors · 2024-03-25
> **Catastrophic Forgetting vs Concept Forgetting (contd.)**
>
> **"I can understand that the pre-training dataset is not often accessible. In absence of the pre-training dataset, the obvious choice would be to set aside a proxy dataset on which the effect of finetuning can be observed. And precisely this is done by the authors (the use of a target dataset). But why does having a target dataset in absence of the pretraining dataset need to rename catastrophic forgetting to concept forgetting? It is still forgetting and frankly speaking, if one does not have the original pretraining dataset, there is no way to tell precisely if the 'concepts' are forgotten or not. This is because, what all concepts the foundation model learnt in the first place can't be known or tested. So, in brief, why the term 'concept forgetting' is being differentiated with 'catastrophic forgetting' is not clear."**
>
> * We are glad that the choice to use a proxy dataset to measure forgetting for a foundation model seems obvious to you. We also agree that there is little else one can do to quantify forgetting on a model for which we don’t know the full extent of the model’s pre-trained knowledge.
> * As mentioned above and in the paper, we don’t claim conceptual novelty on the term concept forgetting, neither do we claim that this is a new phenomenon we study first in this paper.
> * The purpose of this term is simply to identify the setting in which we are measuring forgetting, i.e., for a foundation model whose pre-trained knowledge base is massive but obscure and is being fine-tuned on a certain set of downstream concepts, thereby leading to forgetting of other pre-trained concepts.
>
> **"Incremental Novelty: The paper proposes an approach similar to knowledge distillation which has been explored before. LwF [10 in main paper] and ICARL [11 in main paper] are two classical continual learning papers which have used such approaches. While the authors apply the feature regularisation over all layers as compared to final layers by knowledge distillation, the novelty seems incremental."**
>
> * We have compared with both LwF and iCARL, along with 3 other continual learning baselines in Table 3 of the main paper.
> * We also provide a detailed discussion on distillation methods for continual learning, how LDIFS is different from them and how the use of earlier layers of the network for distillation is crucial to LDIFS’s performance. Please see Section 6 for details.

---

> ### Author Response · Authors · 2024-03-25
> **More results & Questions on claim**
>
> **"More results: Section 5: The authors showed comparisons on similar Continual Learning (CL) Datasets but not the same ones. Why not use the same datasets as is generally followed in CL literature - say 5-datasets or ImageNet-R or CUB etc. Additionally, there exists prior works applying Continual Learning on CLIP models [4, 5, 6, 7 ]. The authors need to compare with them."**
>
> * As mentioned above, our motivation was not to improve performance on classic continual learning or class-incremental learning benchmarks. Hence, we evaluated using a different setup.
> * Having said that, we can apply LDIFS in class-incremental benchmarks. We provide an experiment on Split ImageNet-R comparing LP-init-LDIFS with L2P [2], DualPrompt [3], Coda-Prompt [1], Continual CLIP [5] and SLCA [8]. These are the papers you suggested for which we found code available online and were able to perform comparisons within the limited time-period. For each baseline, we used the CLIP ViT-B/16 as the pre-trained encoder. Results are as follows.
>
> | Method               | Average accuracy  |
> |----------------------|-------------------|
> | L2P [2]              | 74.6 +- 1.21      |
> | DualPrompt [3]       | 77.24 +- 1.27     |
> | Coda-Prompt [1]      | 78.13 +- 1.18     |
> | Continual-CLIP [5]   | 76.23 +- 1.18     |
> | SLCA [8]             | 81.22 +- 1.23     |
> | LP-init-LDIFS (Ours) | **83.62 +- 1.16** |
>
> **Observation**: LP-init-LDIFS outperforms competitive baselines indicating its usability in classic class-incremental learning benchmarks as well.
>
> **"Doubts on Claim: The claim that full fine-tuning can cripple your foundation model is somewhat misleading as works like SLCA[8] and FSA[9] has proven otherwise. The authors need to show results with these approaches as well."**
>
> * The two papers, SLCA [8] and FSA [9] don’t dispute or cast doubt on our claim. Rather, these works validate our claim that classic end-to-end fine-tuning of pre-trained models can lead to forgetting and hence, they propose alternatives which mitigate the issue.
> * **SLCA (Slow Learning with Classifier Alignment)**: the proposed method uses different learning rates for the backbone/encoder and the linear head. The idea is to use a much smaller learning rate for the encoder which can be linked with trying to preserve the pre-trained knowledge of the encoder.
> * **FSA (First Session Adaptation)**: the encoder is only adapted in the first session following which it is frozen, thereby again limiting the amount of change a pre-trained encoder goes through during adaptation. Furthermore, in the few-shot setting, specialized adapter modules called Featurewise-Linear Modulation adapters are used and in the high-shot setting, full body adaptation is used.
> * Thus, both these works develop alternative fine-tuning strategies and recipes to mitigate forgetting.
> * Finally, we have compared with SLCA on Split-ImageNet-R and provide results above.

---

### Review · Reviewer_zv3Y · 2024-03-21

**Summary Of Contributions:**

This paper investigates "concept forgetting" in foundation models, where fine-tuning on specific tasks degrades the model's ability to recognize concepts outside those tasks. It introduces a fine-tuning method, LDIFS (ℓ2 distance in feature space), designed to preserve pre-trained knowledge while learning new concepts. LDIFS minimizes the feature space distance between the original and fine-tuned models, significantly reducing concept forgetting. Extensive experiments across 10 tasks demonstrate LDIFS's effectiveness in mitigating concept forgetting compared to traditional methods. Additionally, in a continual fine-tuning setup, LDIFS shows promise in minimizing forgetting and facilitating positive forward transfer, where learning new tasks enhances performance on previously learned ones. The study highlights the importance of maintaining foundational knowledge in fine-tuned models and presents LDIFS as a viable solution to concept forgetting, marking a significant contribution to the field.

**Audience:**

Yes

**Claims And Evidence:**

Yes

**Requested Changes:**

1) Either focus more on CLIP and motivate the paper or conduct experiments of variety of foundation models

2) The long term learning impact of continual learning. If extensive sequences are used then would there be cumulative biases or errors introduced? (Nice to have)

3) A discussion on weaknesses of LDIFS

4) Comparison with commonly used data training techniques such as joint training with pre/finetuning data.

**Strengths And Weaknesses:**

Pros:
1) The paper is well written and easy to follow.
2) The experiments with CLIP ViT-B/32 are thorough with a variety of downstream tasks and results.
3) Paper presents an intuitive and easy to follow LDIFS  concept with thorough discussion on the method technique.


Cons:
1) The motivation of usage of LDIFS models is not strong, paper goes deep in CLIP model but wants to generalise findings to other foundational models without evidence.
2) Authors need to provide more evidence to support full fine-tuning can cripple your foundation models. There are methods such as joint training and others as provided in the comments below by fellow reviewers. How do those methods compare with LDIFS? Can we achieve similar results?
3) The paper does not go in depth of weaknesses of  LDFIS.Where and when could it hinder performance? Is that dependent on model size or data or other parameters?

---

> ### Author Response · Authors · 2024-03-25
> **Model ablations, comparison with joint training, limitations of LDIFS**
>
> Thank you for taking the time to review our work and for your feedback! We provide our responses below.
>
> **”The motivation of usage of LDIFS models is not strong, paper goes deep in CLIP model but wants to generalise findings to other foundational models without evidence“**
>
> **Either focus more on CLIP and motivate the paper or conduct experiments of variety of foundation models**
>
> Thank you for this comment.
> * We have performed an extensive ablation across both different available pre-trained architectures of CLIP as well as other image-text foundation models like FLAVA and vision only models like DINOv2 and MAE, on fine-tuning on ImageNet.
> * For details, please see our general response: “Concept forgetting beyond CLIP ViT-B/32”.
> * We found concept forgetting to exist across model types (CLIP, FLAVA, DINOv2 and MAE) as well as architectures (ViT-B, ViT-L).
> * Our proposed baseline ***LP-init-LDIFS consistently outperformed other competitive fine-tuning baselines on all these models***.
>
> **”Authors need to provide more evidence to support full fine-tuning can cripple your foundation models. There are methods such as joint training and others as provided in the comments below by fellow reviewers. How do those methods compare with LDIFS? Can we achieve similar results?”**
> **Comparison with commonly used data training techniques such as joint training with pre/finetuning data.**
>
> Thank you for this comment.
> * We compared LP-init-LDIFS with the joint training baseline where we fine-tune on the full training set of all datasets seen previously in the sequence along with the dataset of the current task in the sequence.
> * For details, please see our general response: “Comparing LP-init-LDIFS with joint training”. Our observations are as follows:
> * LP-init-LDIFS is competitive but slightly underperforms the joint training baseline on previous tasks in the sequence. Note that joint training sees the full training data of previous tasks as well.
> * LP-init-LDIFS is competitive with joint training on the current/last task in the sequence.
> * LP-init-LDIFS significantly outperforms joint training on preserving pre-trained performance on tasks which aren’t part of the sequence and we don’t fine-tune on.
>
> **"The paper does not go in depth of weaknesses of LDFIS.Where and when could it hinder performance? Is that dependent on model size or data or other parameters?"**
>
> **"A discussion on weaknesses of LDIFS"**
>
> Thank you for bringing up this point. In what follows, we discuss some of the limitations of the LDIFS method, which we identified in this work.
>
> * The LDIFS regularizer has an associated hyperparameter $\lambda_{\mathrm{LDIFS}}$ which balances the objective of preserving pre-trained model behaviour (i.e., minimizing feature space distance with pre-trained model) with the objective of fine-tuning on the task at hand. Following convention, we cross-validate this hyperparameter on a held-out validation set from the task we fine-tune on. While this performs well in our evaluations, it is interesting to see if there can be better ways of setting this hyperparameter and in which cases the cross-validated hyperparameter value may not yield the best model either at concept forgetting or fine-tuned performance.
> * To compute LDIFS, we use a fixed set of features evenly spaced out in the network. We can possibly improve performance further if we know which features are useful for encoding which concepts. Work in this direction could also provide interesting insights and increase our understanding on how concepts are encoded in the feature space of models.
> * LDIFS tries to preserve the behaviour of the pre-trained model while fine-tuning. If there are biases and undesirable behaviours in the pre-trained model, these can percolate into the fine-tuned model. It is therefore worth studying how such undesirable behaviours can be prevented from being distilled into the fine-tuned model while preserving the pre-trained knowledge.
>
> Regarding the performance of LDIFS with varying model size, as seen from our model ablations experiment, LDIFS works well both with smaller models like ResNet-50 as well as larger ones like ViT-L. We have updated the Conclusions & Remarks (Section 7 of the main paper) to include this discussion as Limitations and Future work.

---

> > ### Author Response · Authors · 2024-03-25
> > **Long term impact of continual learning**
> >
> > **”The long term learning impact of continual learning. If extensive sequences are used then would there be cumulative biases or errors introduced? (Nice to have)”**
> >
> > Thank you for this point.
> > * It is certainly interesting to see how fine-tuning with LDIFS over long sequences can introduce biases or errors over the sequence.
> > * Additionally, in our work, we have some observations on how changing the order of the sequence can affect the overall performance of continual fine-tuning with features learnt from some tasks being helpful for others but not necessarily the other way around (see Section 4 second last para “Positive Forward Transfer”).
> > * There exist longer sequence benchmarks like NEVIS’22 [R4] on which evaluating LDIFS and other fine-tuning methods can reveal interesting insights. However, performing such analysis on longer sequences is something we are looking to study as future work.
> >
> > [R4] Bornschein, J., Galashov, A., Hemsley, R., Rannen-Triki, A., Chen, Y., Chaudhry, A., He, X.O., Douillard, A., Caccia, M., Feng, Q. and Shen, J., 2023. Nevis' 22: A Stream of 100 Tasks Sampled from 30 Years of Computer Vision Research. Journal of Machine Learning Research, 24(308), pp.1-77.

---

### Author Response · Authors · 2024-03-25
**Concept forgetting beyond CLIP ViT-B/32**

We have performed a thorough model ablation to investigate if LDIFS is able to minimize concept forgetting in models other than a CLIP ViT-B/32.

**Models**
* Within CLIP, we use ResNet-50, ViT-B/16 and ViT-L/14 in addition to ViT-B/32.
* We also introduce a FLAVA [R1] ViT-B/16 model as another example of a multi-modal (image-text) foundation model.
* Finally, we experiment with two vision-only pre-trained foundation models: DINOv2 [R2] and Masked Auto-Encoder (MAE) [R3]. For DINOv2, we use ViT-B/14 and ViT-L/14 and for MAE, we use the ViT-B/16 and ViT-L/16 architectures.

**Evaluation setup**
* We fine-tune all the models on ImageNet and measure forgetting on a total of 5 datasets: DTD, EuroSAT, GTSRB, RESISC45 and SVHN.
* Similar to Table 1 in the main paper, we report $A_{\mathrm{LP}}$ on ImageNet, which quantifies fine-tuned performance on ImageNet, along with average $\Delta_{\mathrm{LP}}$ on all the other 5 datasets which quantifies concept forgetting.
* For vision-only models, DINOv2 and MAE, FLYP and FLYP-CE fine-tuning baselines are not applicable since they rely specifically on text encoder representations as well. Furthermore, for these models, we use a randomly initialized linear head in ZS-init fine-tuning.

**Results**
* We present results in the table below. We have shown the best $\Delta_{\mathrm{LP}}$ values in bold.

| Model  | Architecture | FLYP A_LP | FLYP Del_LP | FLYP_CE A_LP | FLYP_CE Del_LP | ZS_CE A_LP | ZS_CE Del_LP | LP_CE A_LP | LP_CE Del_LP | ZS_L2SP A_LP | ZS_L2SP Del_LP | LP_L2SP A_LP | LP_L2SP Del_LP | ZS_LDIFS A_LP | ZS_LDIFS Del_LP | LP_LDIFS A_LP | LP_LDIFS Del_LP |
|--------|--------------|-----------|-------------|--------------|----------------|------------|--------------|------------|--------------|--------------|----------------|--------------|----------------|---------------|-----------------|---------------|-----------------|
| CLIP   | ResNet-50    | 78.64     | -4.18       | 78.42        | -3.92          | 78.39      | -4.01        | 78.45      | -3.4         | 76.2         | -2.1           | 76.13        | -1.54          | 77.94         | -0.67           | 78.16         | **-0.11**           |
| CLIP   | ViT-B/32     | 82.26     | -3.74       | 82.18        | -2.46          | 82.02      | -3.02        | 82.12      | -2.17        | 80.9         | -1.13          | 80.78        | -0.88          | 82.14         | 0.02            | 82.21         | **0.1**             |
| CLIP   | ViT-B/16     | 85.6      | -2.87       | 85.38        | -2.16          | 85.21      | -2.92        | 85.36      | -1.73        | 83.11        | -1.03          | 82.19        | -0.74          | 85.24         | 0.07            | 85.31         | **0.16**            |
| CLIP   | ViT-L/14     | 88.01     | -2.4        | 87.96        | -1.6           | 87.88      | -2.33        | 87.91      | -1.52        | 86.94        | -0.87          | 86.87        | -0.43          | 87.74         | 0.12            | 87.85         | **0.22**            |
| FLAVA  | ViT-B/16     | 81.79     | -4.07       | 81.54        | -3.72          | 81.18      | -3.94        | 81.36      | -3.04        | 79.67        | -1.82          | 80.11        | -1.1           | 81.24         | -0.34           | 81.61         | **0.04**            |
| DINOv2 | ViT-B/14     | -         | -           | -            | -              | 85.32      | -2.71        | 85.48      | -1.86        | 84.16        | -0.92          | 84.5         | -0.66          | 85.63         | 0.01            | 86.02         | **0.06**            |
| DINOv2 | ViT-L/14     | -         | -           | -            | -              | 87.6       | -1.92        | 87.9       | -1.4         | 86.63        | -0.64          | 87.02        | -0.19          | 87.84         | 0.08            | 87.91         | **0.13**            |
| MAE    | ViT-B/16     | -         | -           | -            | -              | 83.57      | -5.1         | 83.81      | -4.36        | 82.7         | -3.87          | 82.84        | -3.03          | 83.39         | -1.62           | 83.76         | **-0.94**           |
| MAE    | ViT-L/16     | -         | -           | -            | -              | 85.86      | -4.26        | 86.04      | -3.59        | 85.02        | -2.64          | 85.1         | -1.82          | 85.47         | -0.75           | 85.9          | **-0.12**           |

Our observations and changes to the paper are detailed below.

---

> ### Author Response · Authors · 2024-03-25
> **Concept forgetting beyond CLIP ViT-B/32 contd.**
>
> **Observations**
> * **Concept forgetting in other models**: For fine-tuning baselines other than LDIFS, we observe a consistently negative $\Delta_{\mathrm{LP}}$ across models and architectures indicating the presence of concept forgetting for all these models.
>
> * **Large models forget less**: Within model types, ViT-L architectures exhibit lesser concept forgetting than their ViT-B counterparts indicating that larger models may tend to forget pre-trained knowledge less.
>
> * **CLIP and DINOv2 are competitive and outperform FLAVA and MAE**: The CLIP and DINOv2 models are better performers in terms of minimizing concept forgetting. However, it is not clear if this is due to larger pre-training sets or is an artefact of the self-supervised learning method. This requires further investigation for future work.
>
> * **LP-init-LDIFS is a consistent winner in minimizing concept forgetting**: Across all model types and architectures, we observe LP-init-LDIFS to consistently have the highest, mostly positive $\Delta_{\mathrm{LP}}$ values indicating not only minimized concept forgetting but also positive forward transfer in many cases.
>
>
> **Changes to revised paper draft**
>
> We have put this entire experiment in Appendix C.5 of the updated draft of the paper.
>
>
> [R1] Singh, A., Hu, R., Goswami, V., Couairon, G., Galuba, W., Rohrbach, M. and Kiela, D., 2022. Flava: A foundational language and vision alignment model. In Proceedings of the IEEE/CVF Conference on Computer Vision and Pattern Recognition (pp. 15638-15650).
>
> [R2] Oquab, M., Darcet, T., Moutakanni, T., Vo, H., Szafraniec, M., Khalidov, V., Fernandez, P., Haziza, D., Massa, F., El-Nouby, A. and Assran, M., 2023. Dinov2: Learning robust visual features without supervision. arXiv preprint arXiv:2304.07193.
>
> [R3] He, K., Chen, X., Xie, S., Li, Y., Dollár, P. and Girshick, R., 2022. Masked autoencoders are scalable vision learners. In Proceedings of the IEEE/CVF conference on computer vision and pattern recognition (pp. 16000-16009).

---

### Author Response · Authors · 2024-03-25
**Comparing LP-init-LDIFS with joint training**

We performed an experiment comparing LDIFS with joint training in our 3-task continual fine-tuning setup.

**Experiment setup**
* For the joint training baseline, at each point in the sequence, we use the full training data of the datasets seen previously in the sequence along with the training set of the task that is currently being introduced.
* Similar to Table 2 of the main paper, where we report continual fine-tuning results, we report $\Delta_{\mathrm{LP}}$ and $A_{\mathrm{LP}}$ and compare with LP-init-LDIFS.
* We report performance on the sequence of datasets on which we perform fine-tuning (SVHN, CIFAR-10 etc) as well as the average $\Delta_{\mathrm{LP}}$ and $A_{\mathrm{LP}}$ on 6 other datasets (from datasets in Table 1 of the main paper, except ImageNet) on which we don't perform any fine-tuning.
* The goal is to attain fine-tuned performance on the ones which are included in the sequence and to preserve pre-trained performance on the datasets we don't fine-tune on.

**Results**
In the table below, JT stands for joint training.

| Fine-tune sequence  | Eval set | Del_LP JT  | A_LP JT  | Del_LP LDIFS | A_LP LDIFS  |
|---------------------|----------|-----------|---------|-----------------|---------------|
| SVHN -> C10 -> R45  | SVHN     | -0.02     | 97.04   | -0.41           | 96.68         |
| SVHN -> C10 -> R45  | C10      | -0.1      | 97.76   | -0.21           | 97.41         |
| SVHN -> C10 -> R45  | R45      | 3.88      | 95.19   | 3.84            | 95.0          |
| SVHN -> C10 -> R45  | Others   | -4.74     | 80.13   | 0.1             | 87.08         |
| SVHN -> C100 -> R45 | SVHN     | -0.23     | 96.77   | -0.65           | 96.32         |
| SVHN -> C100 -> R45 | C100     | 0.05      | 87.02   | -0.3            | 86.54         |
| SVHN -> C100 -> R45 | R45      | 3.86      | 95.07   | 3.83            | 95.11         |
| SVHN -> C100 -> R45 | Others   | -4.67     | 80.62   | -0.23           | 89.12         |
| SVHN -> Cars -> R45 | SVHN     | -0.11     | 97.13   | -0.17           | 96.9          |
| SVHN -> Cars -> R45 | Cars     | 0.32      | 84.12   | 0.47            | 84.23         |
| SVHN -> Cars -> R45 | R45      | 3.83      | 95.24   | 3.83            | 95.27         |
| SVHN -> Cars -> R45 | Others   | -4.95     | 83.42   | 0.23            | 89.39         |

**Observations**
* ***Earlier tasks in the sequence:*** On tasks seen earlier in the sequence, (like SVHN), LP-init-LDIFS is competitive but slightly underperforms compared to joint training which sees the full training set comprising all previous tasks at every point in the sequence.
* ***Current/Last task in the sequence:*** On the final task in the sequence, i.e., RESISC45, LDIFS performs competitively with the joint training baseline.
* ***Tasks not in the sequence:*** On tasks not seen in the fine-tuning sequence, LDIFS significantly outperforms joint training exhibiting its ability to minimize concept forgetting.

**Changes to revised draft**
We have included this experiment along with relevant discussion in Appendix C.8 of the revised draft.

---

### Author Response · Authors · 2024-04-03
**Gentle nudge**

Dear reviewers,

Thank you very much for the time you spent reviewing our work! We appreciate the comments you provided and have done our best to address all your concerns, including additional experiments, clarifications and changes in the draft to reflect the same.

In this regard, we request you to kindly have a look at our responses and please let us know if there are any further queries from your end. We believe that we have provided satisfactory responses and would be happy to address further queries if any.

Thank you again for your time and we look forward to your response.

---

### Decision · Action_Editor_cmbG · 2024-05-27

**Recommendation:** Accept with minor revision

**Comment:**

All (but one) reviewers agree that the paper should be accepted. This reviewer's main concern is that the proposed fine-tuning setting is not fully realistic. Still, the reviewer agrees that the "Claims And Evidence" and "Audience" criteria are met.
While I agree this is a small concern, the evaluation and suggested solution are of high interest to the community (this is also a question more of impact, which is not part of the acceptance decision).

In fact, I believe the paper should be considered for Featured Certification, given the breadth of the experiments and potential for interest in the community. The proposed solution is easy to realize and has the potential to show wide applicability, given this is a fundamental problem.

For the minor revision, I would like to see:
Required:
1. Include the appendix as part of the main pdf
2. Include Split-ImageNet-R somewhere (if it is not added yet)
3. Include any aspects of the author's response not integrated so far

Optional:
1. I think the paper would be more impactful if the experiments provided in the author's response / revision of the appendix would not be an afterthought in the appendix but were properly integrated into the main paper, especially  Appendix C.5. I think this would give the paper a chance at the journal to ICLR conference track.

**Audience:**

All reviewers agree that this is of interest to the audience.

I concur as fine-tuning foundation model is the standard practice and of very broad interest in the community.

**Claims And Evidence:**

All reviewers agree that the claims made in the submission are supported appropriately.

In fact, the main claim of the paper, as the title states, "Fine-tuning can cripple your foundation model," and the recommended proposal, "preserving features may be the solution," is well supported by the submission and significantly strengthened by the broadened experiments provided in the author response / latest revision.